# The amygdala instructs insular feedback for affective learning

**Dominic Kargl[1†], Joanna Kaczanowska[1†], Sophia Ulonska[2], Florian Groessl[1], Lukasz Piszczek[1], Jelena Lazovic[3], Katja Buehler[2], Wulf Haubensak[1]\***

[1]Research Institute of Molecular Pathology (IMP), Vienna Biocenter (VBC), Vienna, Austria; [2]VRVis Zentrum für Virtual Reality und Visualisierung Forschungs-GmbH (VRVis), Vienna, Austria; [3]Preclinical Imaging Facility (pcIMAG), Vienna Biocenter Core Facilities (VBCF), Vienna, Austria

**Abstract** Affective responses depend on assigning value to environmental predictors of threat or reward. Neuroanatomically, this affective value is encoded at both cortical and subcortical levels. However, the purpose of this distributed representation across functional hierarchies remains unclear. Using fMRI in mice, we mapped a discrete cortico-limbic loop between insular cortex (IC), central amygdala (CE), and nucleus basalis of Meynert (NBM), which decomposes the affective value of a conditioned stimulus (CS) into its salience and valence components. In IC, learning integrated unconditioned stimulus (US)-evoked bodily states into CS valence. In turn, CS salience in the CE recruited these CS representations bottom-up via the cholinergic NBM. This way, the CE incorporated interoceptive feedback from IC to improve discrimination of CS valence. Consequently, opto-/chemogenetic uncoupling of hierarchical information flow disrupted affective learning and conditioned responding. Dysfunctional interactions in the IC↔CE/NBM network may underlie intolerance to uncertainty, observed in autism and related psychiatric conditions.

**\*For correspondence:**
wulf.haubensak@imp.ac.at

[†]These authors contributed equally to this work

**Competing interests:** The authors declare that no competing interests exist.

## Introduction

Brains learn about environmental predictors to adapt future behavioral choices (*LeDoux, 2000*). For instance, in Pavlovian learning, the brain updates the CS with its predictive value for unconditioned reward or threat events (*Groessl et al., 2018*; *Schultz and Dickinson, 2000*). Previous research has successfully identified regions, neuronal populations, and mechanisms underlying this form of associative learning (*Grewe et al., 2017*; *LeDoux, 2000*). Essentially, Pavlovian learning relies on associating a CS with basic physiological stimuli (unconditioned stimuli, US) that indicate reward or punishment (*Belova et al., 2007*). The interoceptive insular cortex (IC) plays a fundamental role in sensing these stimuli (*Avery et al., 2017*; *Craig, 2002*; *Critchley et al., 2004*; *Livneh et al., 2020*; *Segerdahl et al., 2015*). In this regard, limbic cortices, in particular the IC, are at the apex of sensory integration and thus represent interoceptive models and associated states in their most abstracted form (*Chanes and Barrett, 2016*; *Pezzulo et al., 2018*). Since IC activity is intricately linked to affect (*Dolensek et al., 2020*), these representations may generate CS value from interoception. Interestingly, the human IC couples to the central amygdala (CE) in resting state functional MRI (fMRI) (*Gorka et al., 2018*; *Schultz et al., 2012*), with neurons in both areas acquiring CS responses over the course of Pavlovian learning (*Shabel and Janak, 2009*; *Vincis and Fontanini, 2016*). As the CE serves as a major gate for conditioned behavior (*Goosens and Maren, 2001*; *Haubensak et al., 2010*; *Li et al., 2013*), the IC and CE may constitute components of a dedicated cortico-limbic network for affective decision-making and Pavlovian learning. Indeed, recent studies have established IC and CE circuitry as a hub for encoding and controlling affective states (*Gehrlach et al., 2019*; *Schiff et al., 2018*; *Venniro et al., 2017*). However, how this circuitry integrates these affective

states into CS value for Pavlovian learning and the mechanisms that gate this integration remain unknown.

Given the prominent functional hierarchical organization of cortico-limbic networks in general, these functions might emerge from top-down and bottom-up interactions between IC and CE. Notably, the CE exhibits cytoarchitectural (*McDonald, 1982*) and functional (*Kim et al., 2017*) properties of the striatum, and analogies in hierarchical organization between the motor and limbic system have been recognized (*Barrett and Simmons, 2015*; *Shipp et al., 2013*). Therefore, as in cortico-striatal motor processing (*Turner and Desmurget, 2010*), hierarchical interactions might be essential for affective learning (*Karalis et al., 2016*; *Likhtik et al., 2014*; *Saez et al., 2017*). Importantly, aberrations in hierarchical processing may underlie the affective aspects of conditions like autism, due to dysfunctional network integration (*Hong et al., 2019*).

So, how could hierarchical interactions integrate affective states into CS value and recruit this information to Pavlovian learning? On the one hand, Pavlovian learning theories posit that the updating of CS value is gated, depending on the uncertainty about its affective consequences (*Pearce and Hall, 1980*; *Rescorla and Wagner, 1972*). In the context of IC–CE circuitry, the basal forebrain, in particular the nucleus basalis of Meynert (NBM), is a likely gate, given its established role in modulating cortical arousal and plasticity (*Puckett et al., 2007*). On the other hand, CS value can be constructed from its underlying salience and valence dimensions (*Cooper and Knutson, 2008*; *Kahnt and Tobler, 2017*; *Lin and Nicolelis, 2008*), analogous to affective states (*Calder et al., 2001*). Importantly, signatures of salience and valence are found across both IC and CE (*Shabel and Janak, 2009*; *Uddin, 2015*). We therefore hypothesized that IC, CE, and the NBM constitute a discrete network for Pavlovian learning. Therein, hierarchical interaction between IC and CE assembles interoceptive CS value from salience and valence dimensions, which is internally gated by the NBM.

In general, such emergent functions are difficult to study in isolated cortical and subcortical network elements, so they remain largely uncharted. Therefore, we here mapped the network-wide organization of CS and US features in IC↔CE/NBM circuitry and explored the hierarchical information flow underlying affective associations.

## Results

### IC and CE are functionally coupled and acquire CS information

Given the known anatomical connectivity between IC and CE, we first explored whether the IC and CE also form a discrete functional unit in brain networks. To this end, small animal resting state fMRI emerges as an effective technology for monitoring global brain states and their interactions with local circuitry (*Gozzi et al., 2010*; *Griessner et al., 2018*). Seed-based brain-wide correlation of the IC blood oxygenation level dependent signal in wild-type mice revealed functional coupling of the IC to the CE (*Figure 1Ai top*, n = 4; see *Figure 1—figure supplement 1A, B* for seed placement/correlation matrix). Conversely, CE seed-based analysis showed coupling with the anterior (aIC) and the posterior (pIC) portion of the IC (*Figure 1Ai bottom*). This brain-wide, unbiased approach delineated a network that functionally couples the IC with the CE. Intriguingly, this network includes the NBM as a potential relay between CE and IC (*Figure 1Aii*, *Figure 1—figure supplement 1C*).

These data suggest that the IC↔CE/NBM network could operate as a functional unit. We next set out to deconstruct functional interactions of key elements in this network. The IC can be functionally parcellated into anterior and posterior domains (aIC and pIC) (*Geuter et al., 2017*). In humans, such rostro-caudal gradients correlate with abstract rule learning and cognitive control (*Badre and D'Esposito, 2007*; *Bahlmann et al., 2015*; *Koechlin and Jubault, 2006*). Within CE, somatostatin$^+$ (SST::Cre, CE$^{SST}$), protein kinase C-δ$^+$ (PKCδ::Cre, CE$^{PKCδ}$), and CEm neurons are critical components for affective learning and behavioral gating (*Fadok et al., 2018*; *Haubensak et al., 2010*; *Kim et al., 2017*; *Li et al., 2013*). Taken together, these individual elements might constitute a hierarchical network encoding Pavlovian stimuli to control conditioned responding.

To access the IC elements in behaving animals, extracellular recordings are well suited due to its anatomical position (particularly the aIC portion of the IC, which is rather inaccessible with other methods). Using this technology, we could sample from 113 neurons in aIC (n = 6 mice) and 98 neurons in pIC (n = 7 mice) per session (*Figure 1B top*, *Figure 1—figure supplement 2*). Conversely,

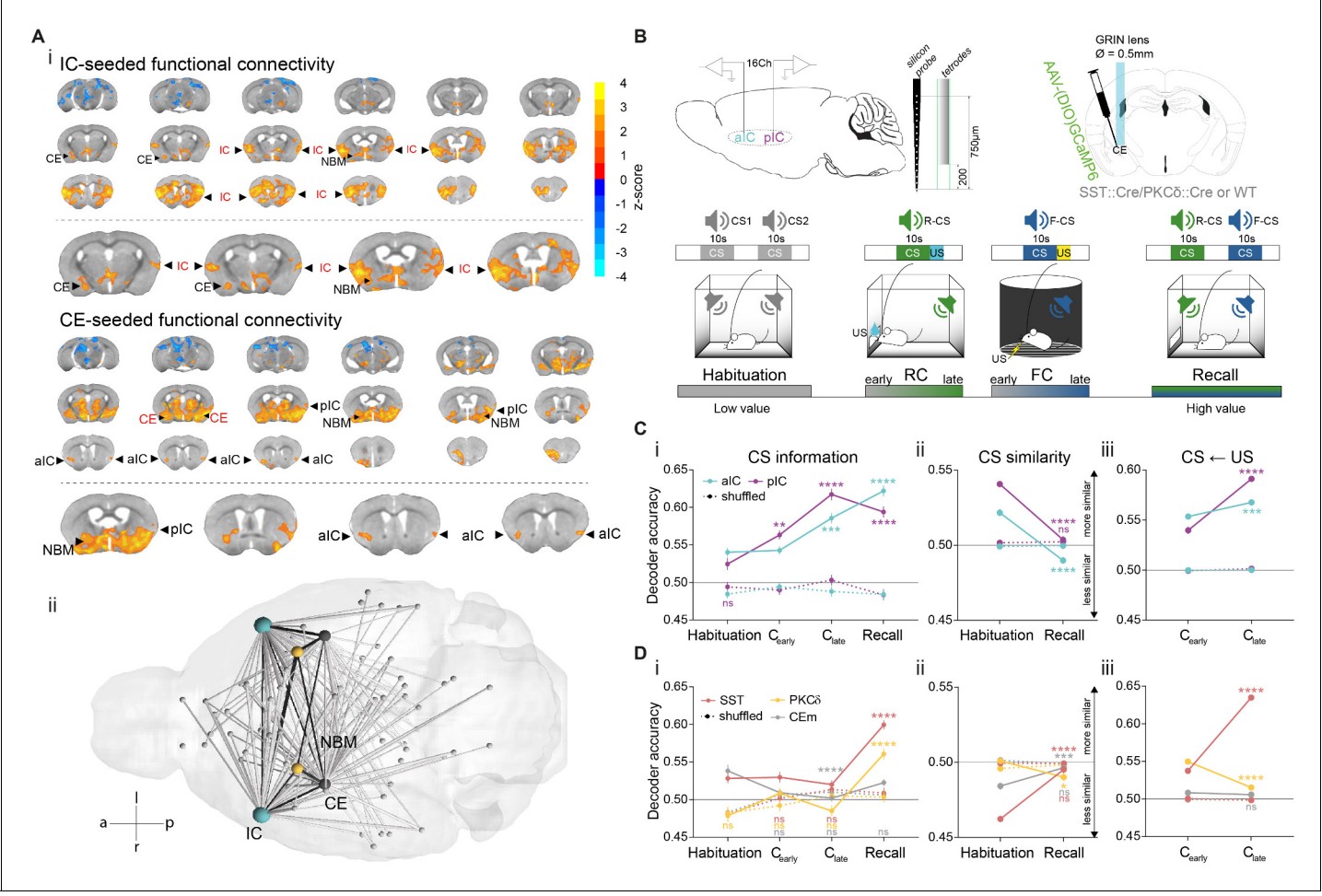

**Figure 1.** IC and CE are coupled and acquire information on task stimuli. (A) (i) Seed-based functional connectivity of the bilateral IC (*top*) showed coupling to CE and NBM. Seeding the CE (*bottom*) showed coupling to the aIC, pIC and NBM (radiological view). Significant z-scored correlations to seed nodes are displayed in orange (positive) and blue (negative). (ii) Bottom-view of region-based functional connectivity of ROIs (see correlation matrix in *Figure 1—figure supplement 1B*). Edge thickness depicts connectivity strength. Only nodes and edges with significant correlations to IC and CE are shown. Edges between IC, CE, and NBM are highlighted in black. (B) Schematic depiction of experimental recordings. *Top, left:* mice were chronically implanted with single-site silicon or multi-site tetrode probes in aIC and pIC. *Top, right:* SST::Cre, PKCδ::Cre, or wild-type mice were chronically implanted with a GRIN lens above CE in animals injected with AAVs carrying GCaMP6. *Bottom:* Experimental timeline of the four-stage discriminatory Pavlovian learning paradigm. (C) (i) Decoder accuracy (Da) of a multi-layer perceptron (MLP) classifier trained to detect CS information in the activity of 200 random draws of 40 neurons per IC subregion for each CS and stage. Mean of both CSs is shown (significant stage x subregion interaction in a two-way ANOVA $F_{9,6384}$=13.69, p<0.0001). * Indicates significant differences from the respective habituation stage. (ii) MLP, trained on 400 random draws of neurons as in (i), to detect R(F)-CS, but applied on F(R)-CS within the habituation and recall stages (significant stage x subregion interaction in a two-way ANOVA, $F_{3,6392}$=42.10, p<0.0001). * Indicates significant differences from the habituation stage. (iii) Mean Da of an MLP trained on the activity of 400 random draws of 40 neurons per IC subregion to detect R-US or F-US applied on R-CS or F-CS, respectively, within the $C_{early}$ and $C_{late}$ stages (significant stage x subregion interaction in a two-way ANOVA $F_{3,6392}$=50.14, p<0.0001). * Indicates significant differences from the $C_{early}$ stage. (D) (i) Da of an MLP trained to detect CS information in the activity of 200 random draws of neurons for each CE population (30 neurons for each $CE^{SST}$ and $CE^{PKCδ}$ and seven neurons for CEm), CS and stage. Mean of both CSs is shown (significant stage x population interaction in a two-way ANOVA $F_{15,9576}$=9.30, p<0.0001). * Indicates significance as in Ci. (ii) MLP, trained on 400 random draws of neurons as in (i), to detect R(F)-CS, but applied on F(R)-CS within the habituation and recall stages (significant stage x population interaction in a two-way ANOVA, $F_{5,9588}$=30.40, p<0.0001). (iii) Mean Da of an MLP trained on the activity of 400 random draws of neurons (30 neurons each for $CE^{SST}$ and $CE^{PKCδ}$, and 10 from CEm) to detect R-US or F-US and applied on R-CS or F-CS, respectively, within the $C_{early}$ and $C_{late}$ stages (significant stage x population interaction in a two-way ANOVA $F_{5,9588}$=339.60, p<0.0001). Holm-Sidak post hoc for all analyses, ****p<0.0001. Only non-significant differences to shuffled data are explicitly indicated ('ns'). All data presented as mean ± SEM. Full statistical report in *Appendix 1—table 1*.

The online version of this article includes the following source data and figure supplement(s) for figure 1:

**Source data 1.** Decoding accuracy of an MLP classifier on iterative draws of neurons from IC and CE populations.

**Figure supplement 1.** ROI-based functional connectivity of the control group.

*Figure 1 continued on next page*

*Figure 1 continued*

**Figure supplement 1—source data 1.** fMRI cross-correlation matrix of CE$_{SHAM}$ in *Figure 1—figure supplement 1B*.
**Figure supplement 2.** Raw data and histological assessment of IC recordings.
**Figure supplement 3.** Histology of calcium imaging cohorts.
**Figure supplement 4.** Representative FOVs and activity from CE calcium imaging.
**Figure supplement 5.** Responses to Pavlovian learning task stimuli in IC.
**Figure supplement 6.** Responses to Pavlovian learning task stimuli in CE.
**Figure supplement 7.** Valence-specific mapping of CS and US features in IC–CE circuitry.
**Figure supplement 8.** Performance-dependent responses to Pavlovian learning task stimuli in IC and CE.

Ca$^{2+}$ imaging is an efficient technology to record from genetically identified neuronal populations in CE. We thus recorded 48 units in CE$^{SST}$ (n = 4), 54 units in CE$^{PKC\delta}$ (n = 5), and 29 units in CEm (n = 4) per session from the right hemisphere, with genetically encoded calcium indicators GCaMP6f/m (*Figure 1B top*, *Figure 1—figure supplement 3*) and extracted calcium events from calcium traces (*Figure 1—figure supplement 4*). Electrophysiological spikes and calcium events were down-sampled to 1 s bins to streamline analyses of neural activity within and across IC and CE elements.

For Pavlovian learning, mice were water-deprived and subjected to a discriminatory auditory reward-fear Pavlovian learning paradigm. After habituation in Context A, a CS (CS1; 10 s, 50 ms white-noise pips at 0.9 Hz) was paired with an appetitive US (R-US, water reward) in reward conditioning sessions (RC) in Context A (R-CS). The same mice then underwent fear conditioning (FC), which paired a second conditioned stimulus (CS2; 10 s, 3kHz-constant tone) with an aversive US (F-US; foot-shock) in Context B (F-CS), followed by a non-reinforced recall stage in Context A (*Figure 1B bottom*). Importantly, this discriminatory Pavlovian learning approach allowed us to deconstruct stimulus value into its underlying valence and salience components, which is not possible using single-valenced fear/reward-only designs.

We propose that encoding task stimuli (CS, US) across cortico-limbic hierarchies is shaped by associating CS-US contingencies, which gradually assigns value to the CS. Consistent with this idea, we found CS- and US-bound responses in population activity, as well as significant single neuron responses to CS and US in both, the IC (*Figure 1—figure supplement 5*) and CE (*Figure 1—figure supplement 6*) in all Pavlovian learning stages. Because learning links CS and US states, information related to the affective value of CSs should increase with learning. Therefore, we probed for CS information within the IC–CE network by training a classifier to decode R-CS and F-CS across Pavlovian learning stages (see 'Single-region decoding' in Materials and methods). By iteratively drawing random neurons from each population and stage, we found that information on CSs increased in IC and CE over time compared to shuffled data (*Figure 1Ci*, Di for mean decoder accuracy of R- and F-CS within each stage and neuronal population; see *Figure 1—figure supplement 7A* for valence-resolved decoding).

For this information to be meaningful, learning systems should differentiate between R- and F-CS based on their predicted outcomes. We probed this with CS-specific classifiers trained to separate R-CS and F-CS-correlated neuronal activity in IC and CE (see 'Discrimination of neural activity' in Materials and methods). IC subregions, CE$^{SST}$ and CEm discriminated both CSs at habituation, which improved further after conditioning for the IC (lower accuracy – lower similarity, *Figure 1—figure supplement 7Bi, ii*). In contrast, CE$^{PKC\delta}$ did not initially differentiate between CSs, but acquired discrimination after learning (*Figure 1—figure supplement 7Bii*).

In this experimental setting, the decoder trained to discriminate R-CS and F-CS is more tuned to differences in sensory representations, as CSs are discriminated throughout the paradigm. Conversely, decoding R-CS or F-CS from the opposite CS is tuned to shared features among CS representations. This way the decoder becomes more sensitive in the temporal domain and thus primarily reports affective modulation. So, we next trained a classifier on one CS and applied it to the other CS (see 'Similarity of neural activity' in Materials and methods). Indeed, we found that the IC shows overlapping CS representations at habituation, which separate after conditioning at recall for IC and CE$^{PKC\delta}$ (*Figure 1Cii, Dii*).

In associative learning, affective features in CS representations should originate from and mirror US states. Thus, these affective CS features should include bodily responses to the primary US

experience. To test this, we trained a classifier on US responses (*Figure 1—figure supplement 7Ci*) and used it to decode CS-evoked neuronal activity. We found that the IC projected US properties onto the respective CS (higher accuracy – higher similarity, *Figure 1Ciii*), potentially endowing CS representations with value. Conversely, CE subpopulations mapped US properties differentially. While $CE^{SST}$ explicitly transferred US properties onto the CS, US and CS features in $CE^{PKC\delta}$ did not share representations, and CEm remained neutral with learning (*Figure 1Diii*; this pattern is consistent across valences, see valence-resolved transfer in *Figure 1—figure supplement 7Cii*).

Interestingly, IC subregions dissociated the primary valence of both USs, as indicated by differential population responses to R-US and F-US in aIC and pIC (*Figure 1—figure supplement 5B*). This result highlights a positive to negative valence gradient along the IC antero-posterior axis. Importantly, the magnitude of US responses in IC correlates with later task performance at recall (*Figure 1—figure supplement 8*, see 'Neuronal responses to task stimuli' in Appendix 1 for details). Unlike the IC, all CE populations responded to both USs (*Figure 1—figure supplement 6B*), suggesting that US responses in CE alone may not offer valence contrast for US discrimination, and thus are tuned to stimulus salience.

In summary, we propose a model wherein $CE^{SST}$ and CEm differentiate intrinsic CS salience at habituation (*Figure 1Dii*). After learning, these intrinsic differences are overridden by the uniform salience component of CS-US associations (in either valence domain) (*Figure 1Diii*, see *Figure 1—figure supplement 7Cii* for valence-resolved transfer). Importantly, in this model, early CS salience in $CE^{PKC\delta}$ is replaced by CS valence information in later learning stages, driving CS discrimination in CE (*Figure 1Dii* and *Figure 1—figure supplement 7Bii*).

## IC–CE information flow facilitates conditioned responding

The representation of CS salience and valence components are distributed across the IC–CE network. In turn, the exchange of this information may be required for conditioned responding in Pavlovian learning. To characterize such cortico-limbic interactions, we first assessed synaptic connectivity between a/pIC and CEl populations by retrograde tracing (*Figure 2—figure supplement 1A*) and slice electrophysiology (*Figure 2A*, *Figure 2—figure supplement 1B*). We found that aIC and pIC innervate CEl subpopulations symmetrically (92% of $PKC\delta^+$/91% of $SST^+$ neurons responsive to aIC and 100% of $PKC\delta^+/SST^+$ neurons to pIC input) (*Figure 2B* and 'IC–CE circuit architecture' in Appendix 1).

To investigate whether CS information in the IC–CE network is relevant for conditioned responding, we trained a random forest (RF) classifier to assess the performance of the network in the representation of CS-bound behavior in iterative random draws of 100 neurons from IC and CE combined (see 'Multi-region decoding' in Materials and methods). A behavioral episode was considered 'correct' if it occurred during the presentation of the respective CS, and 'incorrect' if it occurred before CS onset. This analysis showed that successful association of CS and behavior was linked to correct trial performance (*Figure 2—figure supplement 2Ai left*; RF-associated feature importance in *right* is projected onto the elements of the network graph - see below). We then probed information exchange between IC and CE by quantifying the transfer entropy (TE) from event-aligned (electrophysiological spike or calcium event) 1s-binned activity centered on the onset of behavioral episodes (port visits for R-CS; freezing onsets for F-CS) (*Figure 2—figure supplement 3A*; *Magrans de Abril et al., 2018*). Stimuli or behaviors evoke a state that is generalizable across individuals within our circuit architecture, which makes this approach feasible (*Lizier et al., 2011*). After exploring TE parameter space by considering all possible neuron pairs within each CS and stage, as well as within and across regions, we applied the peak TE from a 1 s history for all subsequent analyses (*Figure 2—figure supplement 3B*). This analysis revealed significant information transfer from IC to CE for correct behavioral decisions (*Figure 2Ci*). Specifically, a subnetwork-specific transfer from aIC to $CE^{PKC\delta}$ and $CE^{SST}$ indicated correct port visits (*Figure 2Ci* green), while a transfer from pIC to $CE^{SST}$ indicated correct freezing onsets (*Figure 2Ci* blue). This top-down information transfer was absent in incorrect behavioral episodes occurring outside of the CS presentation (*Figure 2Cii*). Taken together, this suggests that the information transfer in the IC↔CE/NBM network is critical for conditioned responding.

To experimentally test the behavioral consequences predicted by TE maps, we subjected a cohort of mice to the Pavlovian learning task while we temporally uncoupled IC from CE. Mice received bilateral injections of adeno-associated virus (AAV) carrying either the optogenetic inhibitor

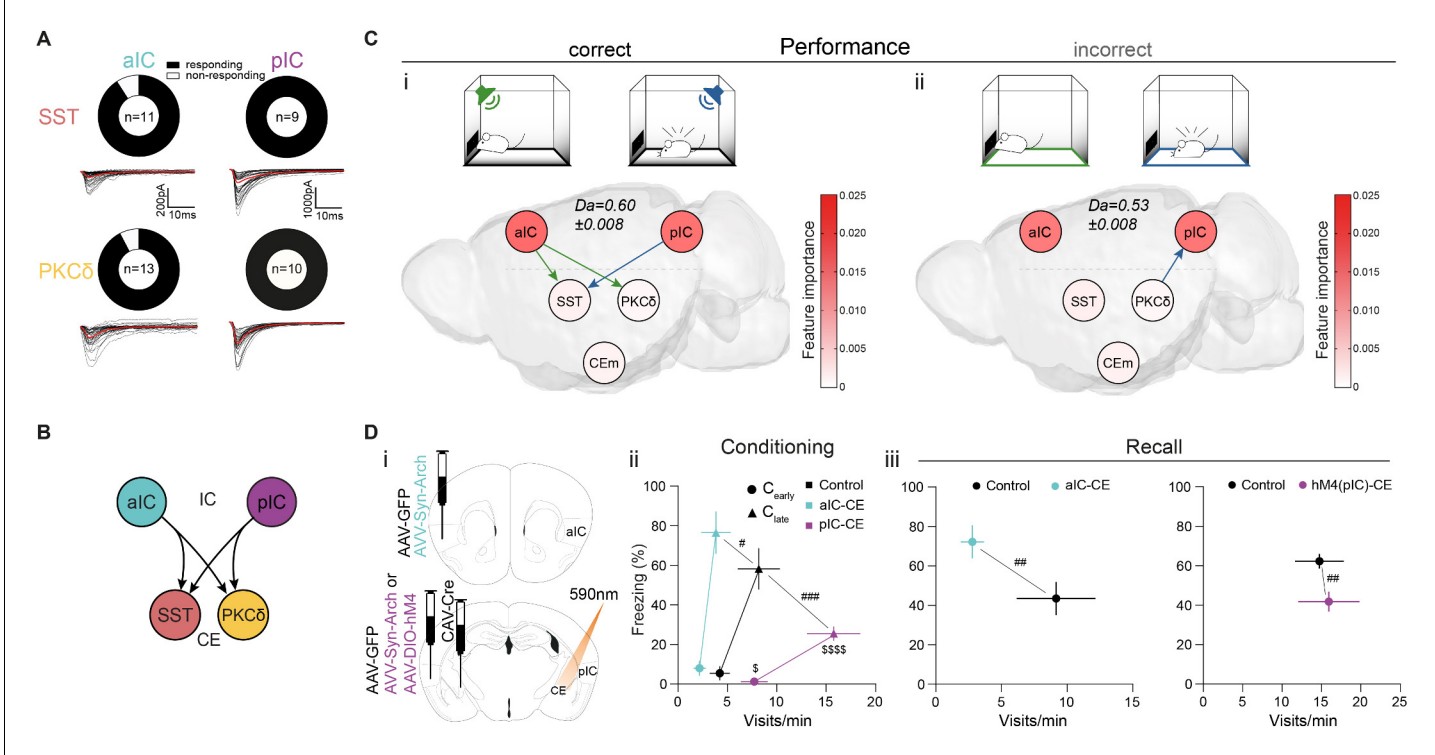

**Figure 2.** IC-CE information flow facilitates conditioned responding. (**A**) Fraction of SST$^+$ and PKCδ$^+$ neurons in CEl that responded with EPSCs upon optogenetic stimulation of aIC or pIC input. (**B**) Scheme for IC inputs to CEl populations. (**C**) Performance-dependent transfer entropy (TE) between IC and CE nodes for (i) correct (port visits during R-CS and freezing episodes during F-CS) and (ii) incorrect (port visits or freezing outside of corresponding CS) behavioral episodes (±2 s of bin containing behavioral episode onset). RF Decoder accuracy (Da) for decoding behavioral episodes shown above networks. Node color corresponds to RF-associated feature importance, indicating information most relevant for RF classification (see *Figure 2—figure supplement 2Ai*). (**D**) (i) Experimental approach to functionally dissect aIC and pIC inputs to CE during a Pavlovian learning task. (ii) Behavioral performance of optogenetic experimental groups in C$_{early}$ and C$_{late}$ stages. Significant MANOVA in C$_{early}$ (F$_{2,44}$=3.60, p=0.0126) and C$_{late}$ (F$_{2,44}$=6.43, p=0.0004). (iii) Behavioral performance of the optogenetic (left) and chemogenetic (hM4(pIC)–CE, right) IC–CE treatment cohorts during manipulation-free recall. Significant MANOVA at recall for the aIC-CE manipulation (F$_{1,13}$=8.18, p=0.005) and pIC-CE manipulation (F$_{1,17}$=6.81, p=0.0067). Data shown as mean ± SEM. n$_{GFP=}$9/12 n$_{aIC–CE=}$7, n$_{pIC–CE=}$9/8. Holm post hoc as difference to control is noted as #, between manipulation groups is noted as $. #/$p<0.05, ##p<0.01, ###p<0.001, $$$$p<0.0001. Full statistical report in *Appendix 1—table 1*.

The online version of this article includes the following source data and figure supplement(s) for figure 2:

**Source data 1.** Approach and avoidance behavior during conditioning and recall in chemogenetic pIC–CE and aIC-pIC manipulation cohorts.

**Figure supplement 1.** Anatomical and functional assessment of IC–CE connectivity.

**Figure supplement 2.** Pavlovian learning task stimulus distributions in the IC–CE network are reshaped by performance and CE$^{PKCδ}$.

**Figure supplement 3.** Illustration for quantifying TE and assessment of TE parameter space.

**Figure supplement 4.** Histological assessment of optogenetic IC–CE manipulation cohorts.

**Figure supplement 5.** Valence-specific conditioned responses of optogenetic IC–CE manipulation cohorts.

**Figure supplement 6.** Histological verification of the chemogenetic pIC–CE and aIC–pIC manipulation cohorts.

**Figure supplement 7.** Valence-specific conditioned responses of chemogenetic pIC–CE and aIC–pIC manipulation cohorts.

archaerhodopsin (syn-Arch) or GFP as control (syn-GFP) into aIC or pIC, and bilateral fiber-optic cannulas placed above CE (*Figure 2Di*; *Figure 2—figure supplement 4*). The respective IC–CE projection was optogenetically inhibited at CS presentations during training. This design specifically interfered with the outflow of CS-associated information from IC to CE. Mice receiving aIC–CE inhibition during CS periods throughout conditioning showed impaired conditioned responding, as indicated by a lower number of port visits in RC and exacerbated freezing in FC compared to control animals (*Figure 2Dii*). In contrast, we observed the opposite pattern for optogenetic pIC–CE manipulation (*Figure 2Dii*; see *Figure 2—figure supplement 5A*).

To test for effects on memory formation, mice underwent a recall session without manipulation. Consistent with the predicted effects of acute silencing, the optogenetic aIC–CE manipulation

interfered with memory acquisition (*Figure 2Diii left*; see *Figure 2—figure supplement 5B* for raw data). Because the acute effects of optogenetic pIC–CE uncoupling did not last into recall, we reasoned that tonic silencing by designer receptor exclusively activated by designer drug (DREADD)-based perturbation of the pIC–CE pathway might be more effective in impacting memory formation in this setting. To achieve this, a separate cohort of wild-type animals received bilateral injections of retrograde canine adenovirus expressing Cre-recombinase (CAV::Cre) into the CE and an AAV for Cre-dependent expression of the inhibitory hM4 receptor bilaterally into the pIC (hM4(pIC)–CE). CAV::Cre in combination with AAV for Cre-dependent GFP expression served as control (*Figure 2—figure supplement 6 top*). The hM4 ligand clozapine-N-oxide (CNO) was systemically administered prior to conditioning sessions. Indeed, tonic pIC–CE silencing at conditioning resulted in a robust impairment of memory formation, as indicated by lower conditioned responding at recall (*Figure 2Diii right*; see *Figure 2—figure supplement 7A, B* for learning curves/raw data).

Collectively, these data demonstrate a functional role for IC–CE interaction in both Pavlovian reward and fear learning, in line with the underlying information flow predicted from TE. We found that the IC innervates CE subpopulations symmetrically, while IC subregions drive conditioned responding antagonistically. Both projections implement Pavlovian memory to adapt behavior for future encounters of sensory cues.

## Learning establishes a performance-linked intra-cortical hierarchy

The previous experiments suggest a link between CS value and behavioral performance. To explore signatures of CS value in the network, we sought to separate CS-driven networks generated from CS periods that lead to a correct behavioral response (port visit during R-CS/freezing episode during F-CS) from CS periods with an incorrect behavioral response (unspecific or absent behavior). This analysis showed that correct conditioned responding is characterized by top-down TE from aIC to pIC (*Figure 3Ai*). These characteristics were different in unsuccessful trials, where TE from aIC to pIC was missing (*Figure 3Aii*). This finding paralleled the observed poorer decoding of CSs not containing correct behavioral episodes, as assessed by RF classification (*Figure 2—figure supplement 2Aii* for RF decoder accuracy and feature importance).

Directional aIC–pIC communication places aIC above pIC in a cortical hierarchy. Top-down processes can ascribe predictions for sensory input to lower elements in the hierarchy, which may facilitate interpretation (*Kok et al., 2014*). To probe for a neurophysiological correlate of an intra-cortical hierarchy in vivo, we simultaneously recorded from aIC and pIC during the Pavlovian learning task (*Figure 3Bi*). We related local spikes (aIC) to distant local field potentials (LFPs in pIC) to assess coherence, which is a proposed mechanism through which neuronal networks exchange information by adjusting gain (*Fries, 2015*). Because performance should scale with learning progress, we chose the best performer in the fear domain at the recall stage (*Figure 1—figure supplement 8A*, 'MS1'). Spike-triggered averages (STAs) of the pIC LFP were generated around spikes from aIC. During habituation, STA amplitudes were similar during CS presentation and a 10 s period immediately preceding CS onset (preCS) (*Figure 3Bii*). Strikingly, during recall, we observed a stimulus-induced increase in STA amplitude, revealing oscillatory synchronization (*Figure 3Biii*). To eliminate potential changes in total LFP amplitudes, we normalized the STA spectrum to the absolute pIC LFP amplitude, yielding spike-field coherence (SFC). During habituation, we observed SFC peaks in the β- and γ-range for preCS, which decreased during CS presentation (*Figure 3Biv*). However, at recall, we observed CS-specific tuning of aIC spikes to pIC LFP, with maximum SFC at 33 Hz (*Figure 3Bv*). SFC was stronger in the negative valence domain, indicating an asymmetry in aIC–pIC communication (*Figure 3—figure supplement 1*). Synchronization was not present in worse performers (*Figure 3—figure supplement 2A*) or when performing the converse analysis (pIC–aIC, *Figure 3—figure supplement 2B*). Taken together, these data reveal stimulus-driven top-down gain modulation within the aIC–pIC network, which correlates with experience and performance.

We then determined the functional relevance of aIC–pIC crosstalk for Pavlovian learning. Animals received bilateral injections of CAV::Cre into the pIC, and an AAV carrying Cre-dependent hM4 (or Cre-dependent GFP for controls) into aIC (*Figure 3Ci*, hM4(aIC)–pIC; *Figure 2—figure supplement 6 bottom*). We systemically administered CNO to both groups at the RC and FC stages and tested their memory during drug-free recall. This specific inhibition of the projection from aIC to pIC during training impaired Pavlovian fear learning (*Figure 3Cii*, see *Figure 2—figure supplement 7Aiii,iv, B*

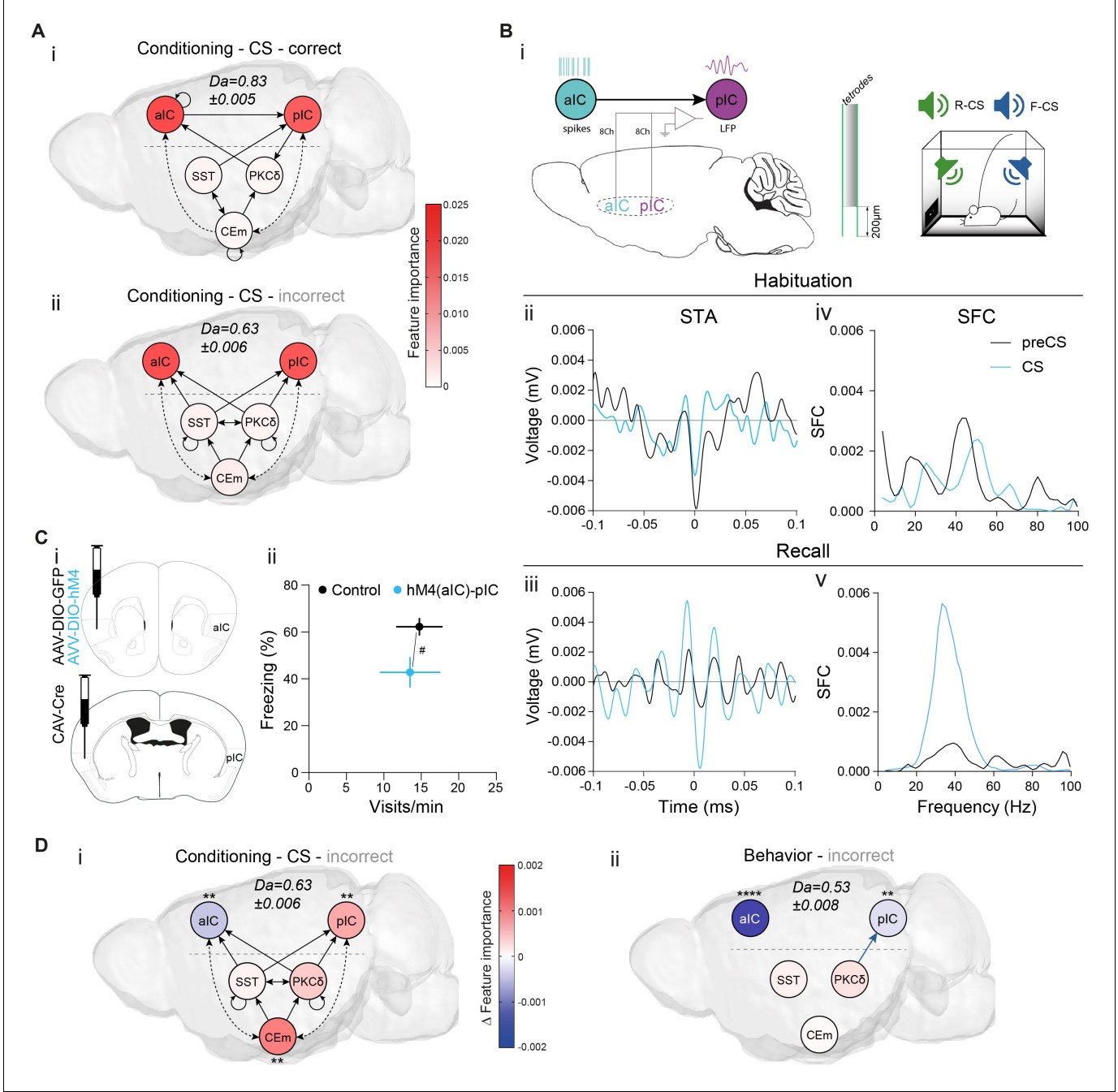

**Figure 3.** Learning establishes a performance-linked intra-cortical hierarchy. (**A**) TE networks generated from CSs during which correct (i) and incorrect/no (ii) behavior occurred during CS presentations. RF decoder accuracy (Da) for decoding correct/incorrect CSs shown above network. Node color corresponds to feature importance from RF classification. (**B**) (i) Scheme of recordings from aIC and pIC multi-site implanted animals, examined for interregional interactions at the habituation and recall stages. (ii, iii) STA from the (ii) habituation and (iii) recall stages of 200 ms pIC LFP traces centered around the occurrence of 2388 preCS/2526 CS (habituation) and 7132 preCS/6920 CS (recall) aIC spikes. (iv, v) pIC LFP power-normalized SFC of STAs for (iv) habituation and (v) recall. (**C**) (i) Experimental strategy for the chemogenetic inhibition of the aIC–pIC pathway. (ii) Quantification of behavioral performance in reward and fear domains at recall with a significant MANOVA ($F_{1,18}=3.64$, p=0.0471), $n_{Controls}=12$, $n_{M4(aIC)-pIC}=9$. Data shown as mean ± SEM. Holm post hoc as difference to control, #p<0.05. Full statistical report in **Appendix 1—table 1**. (**D**) (i) TE network of incorrect CS (from Aii), with node color showing contrast feature importance between incorrect and correct CS. * Depict significantly different feature importance (**Figure 2—figure supplement 2Aii**). RF Da for decoding incorrect CSs shown above network. (ii) TE network (from **Figure 2Cii**) with node color illustrating contrast feature importance between incorrect and correct behavioral episodes. * Depict significantly different feature importance (**Figure 2—figure supplement 2Ai**). RF Da for decoding incorrect behavioral episodes shown above network.

*Figure 3 continued on next page*

*Figure 3 continued*

The online version of this article includes the following source data and figure supplement(s) for figure 3:

**Source data 1.** STA, SFC and associated approach and avoidance behavior in aIC–pIC interaction.
**Figure supplement 1.** The aIC–pIC hierarchy is valence-asymmetric.
**Figure supplement 2.** The aIC–pIC hierarchy is direction-asymmetric and performance-dependent.

for learning curve/raw data). These results provide evidence for top-down gating of associative plasticity in the IC, and support valence-asymmetric gain control established by SFC.

As information flow from aIC is critical for Pavlovian learning, we next tested whether this is also reflected in the distribution of CS- and behavior-related information. We contrasted the feature importance obtained from RF classification between correct/incorrect CSs (*Figure 3A*; *Figure 2—figure supplement 2Aii*) or behavioral episodes (*Figure 2C*; *Figure 2—figure supplement 2Ai*). Indeed, feature importance for decoding in aIC was reduced in incorrect compared to correct CS presentations (*Figure 3Di*), as well as for CS-unspecific behavioral episodes (*Figure 3Dii*). Taken together, these data suggest that CS information in the aIC is critical for Pavlovian learning.

## The basal forebrain mediates bottom-up recruitment of IC activity

Neural systems require mechanisms signaling insufficient CS value to drive learning. To probe for network signatures of insufficient value, we quantified TE between network elements at the time of CS presentation during learning, when only limited CS-US associations have occurred. TE maps during these CS presentations show significant bottom-up transfer from CE to IC, indicating potential recruitment of IC by CE (*Figure 4A*; see 'Control' in *Figure 2—figure supplement 2Aiii* for RF decoder accuracy and feature importance). However, there is no known anatomical projection that could mediate this transfer.

Interestingly, our fMRI survey had identified strong coupling between CE and the cholinergic NBM (*Figure 1Ai bottom*, *Figure 1Aii*). Because electrical stimulation of CE via the basal forebrain (*Kapp et al., 1994*) and activation of putative CE$^{PKC\delta}$ (*Gozzi et al., 2010*) are known to trigger cortical arousal, we hypothesized that the CE–NBM pathway may facilitate IC coupling to CE. The topological organization of NBM projections suggests that distinct subareas innervate specific cortical patches (*Zaborszky et al., 2015*), which could allow NBM inputs to coordinate arousal in selected cortical regions. To investigate this, we made bilateral lesions in CE by injecting N-methyl-D-aspartate (CE$_{NMDA}$, n = 3, *Figure 4—figure supplement 1A*; see *Figure 4—figure supplement 1B* for correlation matrix) to identify regions displaying depleted functional coupling to NBM when compared to CE sham-lesioned control animals (CE$_{SHAM}$). NBM-seeded global brain correlations in the CE$_{NMDA}$ group showed decreased coupling to the right aIC, suggesting that CE input to NBM selectively triggers NBM–aIC interactions (*Figure 4B*; see *Figure 4—figure supplement 1C* for seed placement). To explore this possibility, we assessed synaptic connectivity between CEl populations and NBM neurons by retrograde tracing (*Figure 4—figure supplement 2A*) and slice electrophysiology (see *Figure 4—figure supplement 2B* and 'CE–NBM circuit architecture' in Appendix 1). We found that CEl subpopulations, which are mostly GABAergic (*Cassell et al., 1999*), primarily innervate putatively local parvocellular (pc) interneurons (IN) versus corticopetal magnocellular (mc) neurons, supporting a disinhibitory mechanism of CE input gating NBM output (*Figure 4C*).

To characterize this pathway in vivo, two aIC–pIC multi-site implanted animals (PKCδ::Cre, *Figure 1B*) received an additional injection of an AAV carrying Cre-dependent ChR2 into the right CE, and a fiber-optic cannula placed above the right NBM. This approach directly assessed the effects of CE$^{PKC\delta}$–NBM stimulation on aIC and pIC activity (*Figure 4Di*). Animals received 5 ms 470 nm laser pulses at 0.2 Hz in an open-loop setting while freely moving, which elicited pronounced LFP depolarization in aIC, and, to a lesser extent, in pIC (*Figure 4Dii top/bottom*; comparison of minima of aIC and pIC in *Figure 4Diii*). This stimulation also increased single unit spiking in the IC (*Figure 4—figure supplement 3*), indicating that CE activity may recruit the IC. 405 nm laser pulses served as control stimuli, as ChR2 is insensitive to this wavelength (*Nagel et al., 2003*).

Since the NBM is the major source of acetylcholine in the cortex (*Woolf, 1991*) and CE input may disinhibit choline acetyltransferase$^+$ cholinergic neurons (ChAT$^+$) in the NBM, we asked whether interference with cholinergic signaling could affect IC depolarization. We found that systemic

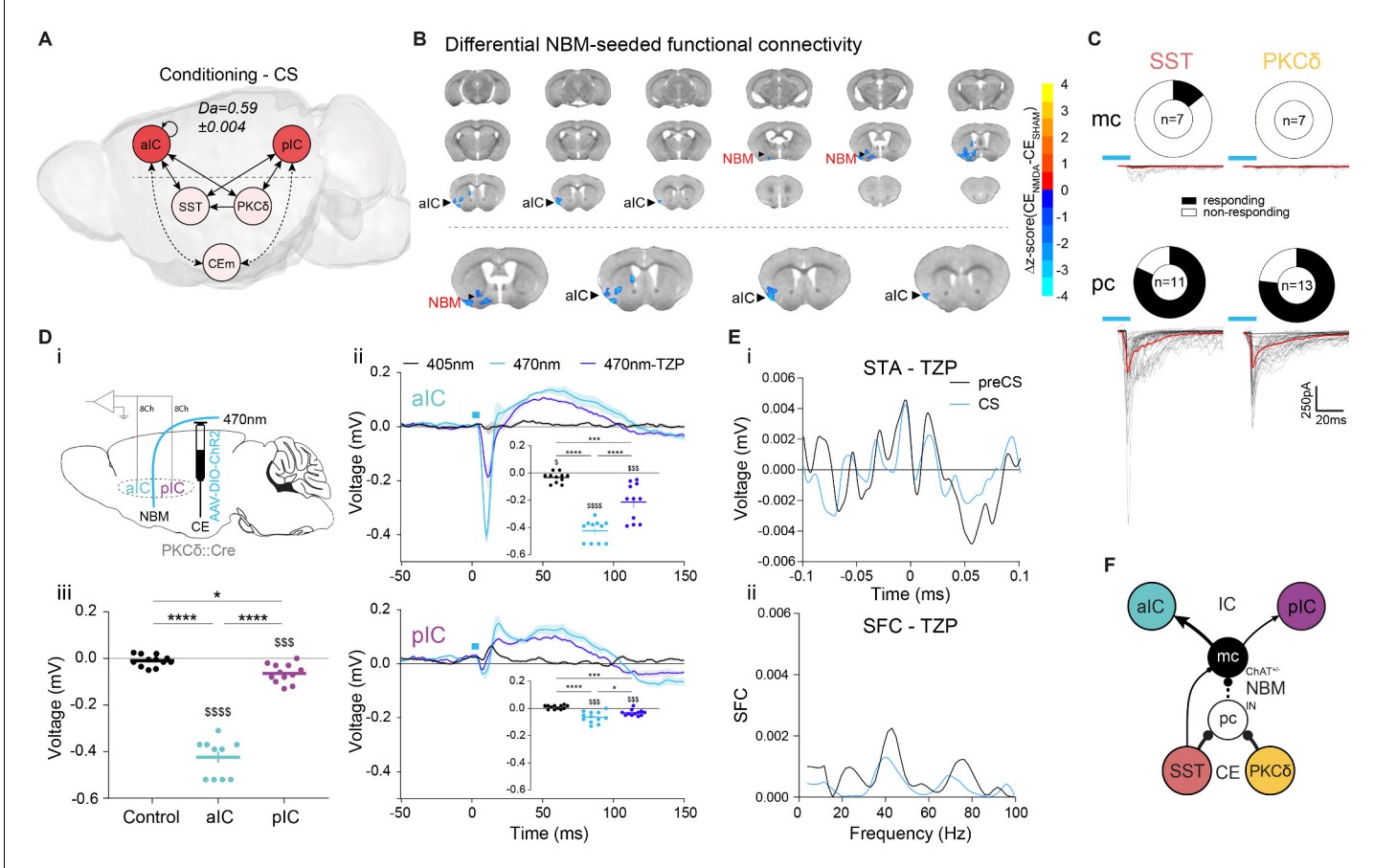

**Figure 4.** Basal forebrain NBM mediates bottom-up recruitment of IC activity. (**A**) (**i**) Network depicting significant TE during CSs generated from data acquired from RC and FC stages. RF decoder accuracy (Da) for decoding CSs at RC and FC stages shown above network. Node color corresponds to feature importance from RF classification (*Figure 2—figure supplement 2Aiii*). (**B**) Chronic $CE_{NMDA}$ reduced NBM resting-state functional connectivity to the right aIC compared to the $CE_{SHAM}$ group. Two-sample t-test between $CE_{SHAM}$ (n = 4) and $CE_{NMDA}$ (n = 3) groups, followed by Gaussian Random Field Theory Multiple Comparison Correction (voxel-level p-value=0.05, cluster-level p-value=0.05). Differential z-score between $CE_{NMDA}$ and $CE_{SHAM}$ indicates depleted correlation (blue). (**C**) Fraction of magnocellular (mc)/parvocellular (pc) neurons in the NBM that responded with IPSCs upon optogenetic stimulation of $CE^{SST}$ or $CE^{PKCδ}$ input. (**D**) (**i**) In vivo optogenetic stimulation of the right $CE^{PKCδ}$–NBM pathway in two IC multi-site recorded, freely moving animals. (**ii**) Peri-laser stimulus time histograms of aIC (top) and pIC (bottom) channel-averaged LFP traces averaged over 60 (405, 470 nm) and 40 (470 nm-TZP) laser pulses. Traces represent averages of all available channels in aIC (11Ch) and pIC (12Ch). Insets depict respective minima of LFP traces within 20 ms after laser pulse onset. Significant one-way RM ANOVA for aIC ($F_{1,116,11,16}$=153.00, p<0.0001) and pIC ($F_{1,340,14,74}$=23.60, p<0.0001). (**iii**) Quantification of IC LFP minima upon $CE^{PKCδ}$–NBM stimulation under control conditions. Significant one-way ANOVA ($F_{2,32}$=209.40, p<0.0001). All data presented as mean ± SEM. Holm-Sidak post hoc analysis was used for comparison between treatments/regions (*) and one-sample t-test for individual differences to zero ($), */$p<0.05, ***/$$$p<0.001, ****/$$$$p<0.0001. Full statistical report in *Appendix 1—table 1*. (**E**) (**i**) STA from the recall stage of 200 ms pIC LFP traces centered around aIC spikes after systemic administration of TZP. (**ii**) SFC resulting from pIC LFP power-normalized STA from (i). (**F**) Circuit model of the bottom-up IC↔CE/NBM pathway consistent with experimental data. Dotted line represents a connection not assessed, but consistent with previous studies (*Jolkkonen et al., 2002*; *Kapp et al., 1994*).

The online version of this article includes the following source data and figure supplement(s) for figure 4:

**Source data 1.** IC LFP responses upon optogenetic $CE^{PKCδ}$–NBM stimulation.

**Figure supplement 1.** ROI-based functional connectivity of the CE lesion group.

**Figure supplement 1—source data 1.** fMRI cross-correlation matrix of $CE_{NMDA}$ in *Figure 4—figure supplement 1B*.

**Figure supplement 2.** Anatomical and functional assessment of CE–NBM connectivity.

**Figure supplement 3.** Optogenetic CE–NBM stimulation increases spiking of single neurons in aIC and pIC.

administration of the muscarinic receptor 1 (M1R) antagonist telenzepine (TZP) dampened $CE^{PKCδ}$–NBM-induced IC depolarization by approximately 50%. These data demonstrate that activity in the CE via NBM interacts with cholinergic modulation of IC function (*Figure 4Dii*).

Because synchronization in the γ-range has been associated with M1R signaling (*Fisahn et al., 2002*), we asked whether it may also be required for intra-IC SFC (*Figure 3B*). To test whether aIC–pIC synchronization is M1R-dependent, we performed recall sessions after systemic administration of TZP. These were interspersed with recall sessions in control conditions (for the same animal) to avoid time effects. We found that M1R antagonism abolished CS-induced SFC, indicating that cholinergic signaling via M1R mediates cortical gain control in the IC (*Figure 4E*).

Collectively, these data support a model whereby CE input to the NBM predominantly inhibits putative GABAergic IN to disinhibit corticopetal ChAT$^{+/-}$ mc neurons (*Figure 4F*). Importantly, these results identify a missing link by which behavioral decisions in the CE may recruit the IC–CE pathway via the NBM (*Gehrlach et al., 2019*; *Venniro et al., 2017*).

## The CE–NBM pathway promotes top-down information for Pavlovian learning

In Pavlovian learning, USs serve as primary prediction error signals to update the CS as a US predictor. TE of the post-US period revealed recurrent dynamics between and within CE populations, as well as bottom-up TE from pIC to aIC. Interestingly, we found bottom-up recruitment of the CE$^{PKCδ}$–aIC pathway, which linked hierarchies during an instructive US (*Figure 5A*). Collectively, an impinging US largely uncoupled the network compared to a CS (*Figure 4A*) and shifted the network TE toward sensory bottom-up signaling (pIC–aIC; see 'Control' in *Figure 2—figure supplement 2Aiv* for RF decoder accuracy and feature importance). To determine whether this phenomenon is solely attributable to primary prediction error, or whether network dynamics represent a general feature of value ambiguity, we examined CS presentations where information on valence was low but relative salience was high. These conditions are best satisfied during habituation, as RF mean decoding accuracy for CS classification was significantly higher compared to conditioning (*Figure 5—figure supplement 1A*). CS-aligned TE networks during habituation were remarkably similar to US-aligned networks at conditioning, suggesting that the CE–NBM–aIC pathway was engaged under conditions of value ambiguity (*Figure 5B*).

To further validate these predictions, we recorded from the IC (as in *Figure 1B*) in mice undergoing conditioning stages when CE$^{PKCδ}$ was chemogenetically silenced. To recalculate TE networks, neural activity from aIC and pIC was replaced with their respective activity from recordings when CE$^{PKCδ}$ was silenced in the same mice (*Figure 5C*; aIC', pIC' in hM4(CE$^{PKCδ}$); *Figure 4A* for control network). In these networks, we still found bottom-up TE from CE to IC. However, recruitment of top-down transfer from IC to CE was absent, reminiscent of TE networks during an incorrectly assigned CS (*Figure 3Aii*). These results indicate that CE$^{PKCδ}$ may be required for IC recruitment. In addition, intra-IC communication displayed pIC to aIC directionality, resembling US/habituation networks (*Figure 5A,B*). This suggests that CE$^{PKCδ}$ activity facilitates top-down information transfer, while sensory bottom-up signaling predominates during CE$^{PKCδ}$ inhibition (*Figure 5C*). Notably, RF CS decoding revealed a shift in feature importance from aIC to pIC (*Figure 5C*, *Figure 2—figure supplement 2Aiii*), which is fully recapitulated in the post-US period (TE: *Figure 5—figure supplement 1Bi*, feature importance: *Figure 2—figure supplement 2Aiv*) and partially recapitulated for CS in habituation (TE: *Figure 5—figure supplement 1Bii*, feature importance: *Figure 2—figure supplement 2Av*).

Ambiguity of CS value evokes bottom-up CE–IC information flow (*Figure 5B*). Because this might be mediated via NBM (*Figure 4*), reducing CE–NBM signaling should interfere with learning. We tested this in a cohort of mice in the Pavlovian learning task with selectively blocked CE$^{PKCδ}$–NBM communication during CS presentations at conditioning. For this, PKCδ::Cre animals were bilaterally injected with Cre-dependent Halorhodopsin or Archaerhodopsin (DIO-NpHR3.0/DIO-Arch) into the CE and implanted with fiber-optic cannulas above NBM (*Figure 5Di*; *Figure 5—figure supplement 2*). Mice receiving optogenetic inhibition of CE$^{PKCδ}$–NBM during all CS periods of conditioning displayed aberrant Pavlovian associations during manipulation-free recall. This was evident from the low number of port visits and reduced freezing levels compared to control animals (*Figure 5Dii*; see *Figure 5—figure supplement 3A, B* for learning curves/raw data). Together, these data reproduce the impaired memory formation observed in aIC– and pIC–CE manipulations (*Figure 2D*). Of note, optogenetic interference with the CE$^{SST}$–NBM pathway had no effect on Pavlovian learning (*Figure 5—figure supplement 2*; *Figure 5—figure supplement 3C, D*).

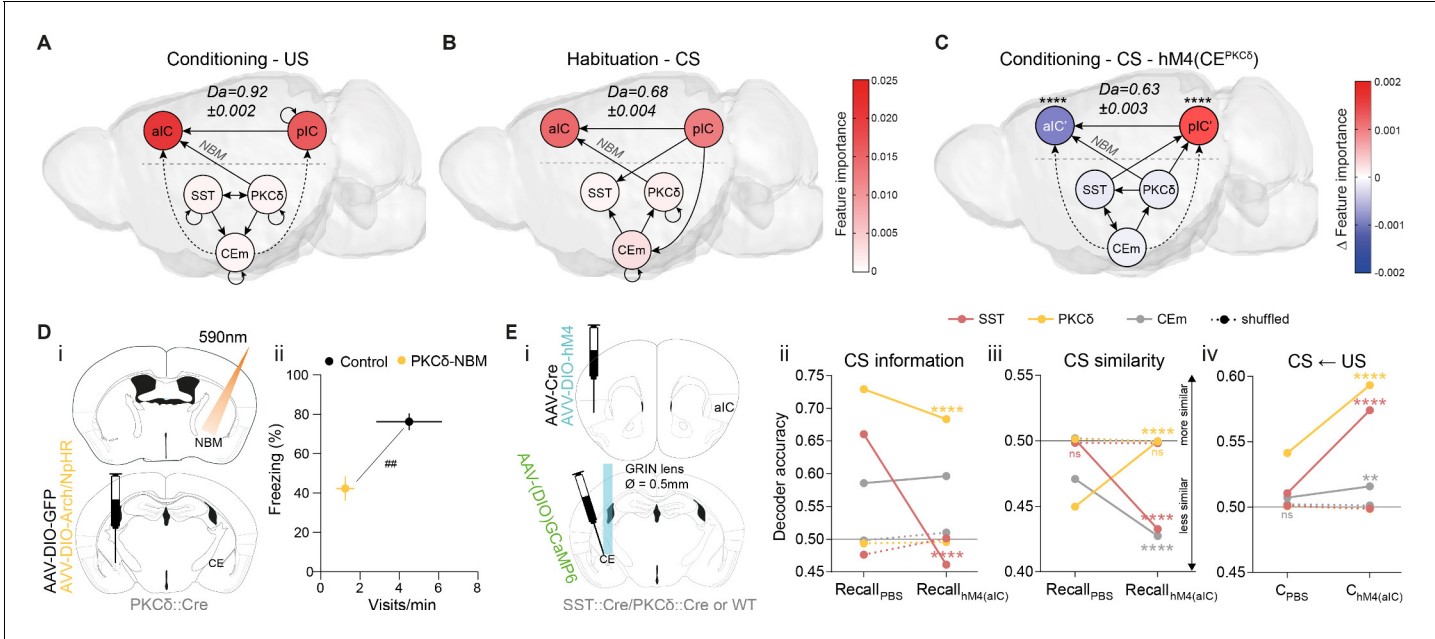

**Figure 5.** The CE–NBM pathway promotes top-down information for Pavlovian learning. (A) Network depicting significant TE after US; generated from data acquired from RC and FC. RF decoder accuracy (Da) for decoding USs at RC and FC stages shown above network. Node color corresponds to feature importance resulting from RF classification under control conditions (see *Figure 2—figure supplement 2Aiv*). (B) Network depicting significant TE during CS in the habituation stage. RF Da for decoding CSs at habituation. Nodes are colored according to the feature importance resulting from RF classification (see *Figure 2—figure supplement 2Av*). (C) Network depicting significant TE during CS generated from data acquired during RC and FC. aIC/pIC data has been replaced by a dataset recorded during chemogenetic inhibition of $CE^{PKC\delta}$ (aIC', pIC') in the same animals (hM4($CE^{PKC\delta}$)). RF Da for decoding CSs at RC and FC stages during hM4($CE^{PKC\delta}$) shown above network. Feature importance given as differential from control conditions, with * indicating significant differences (see *Figure 2—figure supplement 2Aiii*). (D) (i) Experimental approach for optogenetic inhibition of the $CE^{PKC\delta}$-NBM pathway during CS presentations at conditioning. (ii) Quantification of approach and avoidance behavior at recall ($n_{GFP}$ = 7, $n_{CE-PKC\delta-NBM}$=6; significant MANOVA, $F_{1,10}$=9.76, p=0.0045). Data presented as mean ± SEM. Holm post hoc as difference to control, ##p<0.01. (E) (i) Scheme for chemogenetic inhibition of aIC (hM4(aIC)) during CE population recordings. (ii) Mean CS Da of an MLP trained to detect CS information in the activity of 20 $CE^{SST}$, 30 $CE^{PKC\delta}$, and 10 CEm best neurons per treatment to detect CS information at recall during control conditions (PBS) and hM4(aIC) (significant treatment x population interaction in a two-way ANOVA, $F_{5,4788}$=117.50, p<0.0001). (iii) Mean CS Da of an MLP trained on the activity of 20 $CE^{SST}$, 30 $CE^{PKC\delta}$, and 10 CEm best neurons per treatment to detect R(F)-CS applied on F(R)-CS at recall during control conditions (PBS) and hM4(aIC) (significant treatment x population interaction in a two-way ANOVA, $F_{5,9588}$=306.50). (iv) Mean CS Da of an MLP trained on the activity of 20 $CE^{SST}$, 30 $CE^{PKC\delta}$, and 10 CEm best neurons to detect R-US or F-US applied on R-CS or F-CS, respectively, in the conditioning stages during control conditions (PBS) and hM4(aIC) (significant treatment x population interaction in a two-way ANOVA, $F_{5,9588}$=163.90, p<0.0001). * Indicates significant differences between treatments within population, as determined by Holm-Sidak post hoc analysis, ****p<0.0001. Only non-significant differences to shuffled data are explicitly indicated ('ns'). Full statistical report in *Appendix 1—table 1*.

The online version of this article includes the following source data and figure supplement(s) for figure 5:

**Source data 1.** Decoding accuracy of an MLP classifier on single neuron activity of CE populations.
**Source data 2.** Approach and avoidance behavior during the optogenetic $CE^{PKC\delta}$–NBM manipulation cohort during recall.
**Figure supplement 1.** Supporting RF data and TE maps.
**Figure supplement 2.** Histological verification of the optogenetic CE–NBM manipulation cohorts.
**Figure supplement 3.** A cell-type-specific CE–NBM pathway is required for Pavlovian learning.
**Figure supplement 4.** Histology of hM4 expression in calcium imaging cohorts.
**Figure supplement 5.** Valence-specific mapping of CS and US features in IC–CE circuitry.
**Figure supplement 6.** Hierarchical interactions in IC↔CE/NBM circuitry.

IC–CE signaling controls conditioned responding (*Figure 2*), which, in turn, is largely mediated through CE circuitry (*Fadok et al., 2018*). We hypothesized that IC information is critical for the correct representation of CS value in CE (i.e. salience and valence). To test this, we assessed the functional consequences of silencing the aIC. Animals that had been initially used for CE recordings (*Figure 1D*) were used to reassess CS representation and similarity in CE population activity, now having aIC bilaterally silenced ((hM4(aIC)), *Figure 5Ei*; *Figure 5—figure supplement 4*). We focused on neurons most engaged at respective tasks by selecting neurons with the highest decoding

accuracy ('best neurons') using single-neuron decoding (see 'Neural decoding' in Materials and methods), which potentially represented functional ensembles (*Figure 5—figure supplement 5A* and *Figure 5—source data 1*).

Silencing the aIC impaired CS representation in CE$^{SST}$ (*Figure 5Eii*), and CS discrimination by CE$^{PKC\delta}$ to chance level (*Figure 5Eiii* and *Figure 5—figure supplement 5B*). Furthermore, CE$^{SST}$ and CEm reverted to discrimination levels at habituation (see *Figure 1Dii* for comparison). This implies that functionally independent IC pathways channel CS information via CE$^{SST}$ and CS discrimination via CE$^{PKC\delta}$.

Strikingly, aIC silencing revealed a disinhibition of salience transfer from US to CS during conditioning, providing a potential mechanistic explanation for the role of IC–CE pathways in Pavlovian learning (*Figure 5Eiv*; see *Figure 5—figure supplement 5C* for valence-resolved transfer). More specifically, in the absence of aIC function, CE$^{SST}$ and CE$^{PKC\delta}$ map US salience onto CS representations by default, obstructing stimulus discrimination by CE$^{PKC\delta}$. In contrast, successful aIC recruitment confers valence discrimination through CE$^{PKC\delta}$ (*Figure 5Eiii*, *Figure 1Dii* and *Figure 5—figure supplement 5Aii, B*) to guide correct behavioral responding (*Figure 2C*). Collectively, these data demonstrate that reciprocal hierarchical interaction in the cortico-limbic IC↔CE/NBM network ultimately supports salience and valence feature representation in the CE and consequent behavioral decisions (*Figure 2*).

## Discussion

Our study successfully integrated brain wide network analysis from high field small animal fMRI with circuit physiology, and thereby mapped the IC↔CE/NBM network as a distinct functional unit. This approach uncovered a basic functional motif that encodes complementary CS features at different hierarchies and stages of Pavlovian learning. We established a process mechanism, wherein stimulus salience at lower levels recruits top-level value representations in the IC associated with primary reinforcers. This information feeds back to CE to update and reassemble the salience and valence dimensions of the CS to guide behavioral decisions (*Figure 5—figure supplement 6*).

We identified an ascending CE–NBM–IC pathway with a critical role in driving IC-CE signaling. Lesion studies have linked the connection between CE and NBM to enhanced surprise/prediction error-triggered learning (*Han et al., 1999*; *Holland and Gallagher, 2006*). In these settings, the introduction of inconsistency into CS-US contingencies (which increases uncertainty) enhances CS associations and learning, supporting the Pearce-Hall model for Pavlovian learning (*Pearce and Hall, 1980*). In this regard, the CE–NBM pathway could use precision signaling to gate top-down models from higher order areas (aIC) to primary sensory areas (pIC) for sensory learning (*Feldman and Friston, 2010*). This form of striatal coordination of cortical hierarchies, which has been described in humans (*den Ouden et al., 2010*) and may be computationally advantageous (e.g. for gating working memory) (*Frank and Badre, 2012*). In vitro experiments indicate that acetylcholine can favor communication from associative to primary sensory cortex (*Roopun et al., 2010*). Therefore, we speculate that a similar mechanism may gate associative plasticity in the interoceptive system (*Caras and Sanes, 2017*; *Figure 3C*), as acetylcholine has been linked to learning rate and certainty (*Doya, 2002*; *Yu and Dayan, 2005*). Basal forebrain cholinergic neurons rapidly respond to reinforcement feedback in both valence domains (*Hangya et al., 2015*). Since neurons in the CE are unlikely to mediate NBM response to US, we posit that CE neurons and the CE–NBM axis integrate primary reinforcement signals (*Cui et al., 2017*) with information on novelty, confidence, and expectation (*Martinez-Rubio et al., 2018*; *Steinberg et al., 2020*), which is relayed to the IC and the amygdala itself (*Yu et al., 2017*). Indeed, these higher order prediction errors, which incorporate hierarchical probability distributions, have been mapped onto the basal forebrain in humans (*Iglesias et al., 2013*).

Cognitive function requires balanced top-down signaling, while its dysregulation may underlie conditions like autism and schizophrenia (*Friston et al., 2016*; *Lawson et al., 2014*). Disruptions in hierarchical processing (*Figures 3D* and *5C*), analogous to human patients (*Hong et al., 2019*), could account for the absence of affective models in autism and the resulting behavioral difficulty with uncertainty and affective interactions. Since CE–NBM signaling promotes top-down information flow from aIC to pIC, we propose that disrupted functional connectivity in the IC↔CE/NBM network likely contributes to these conditions. Such hierarchical dysfunction may cause the inability to resolve

uncertainty (*Figure 5D*), as seen in autism (*Vasa et al., 2018*) and comorbid anxiety (*Simonoff et al., 2008*). Individuals diagnosed with autism rely less on prior beliefs, suggesting that they may predominantly utilize sensory bottom-up signaling for perception (*Lawson et al., 2017*). This increased sensory bottom-up processing may result from deficits in model-building and reflect augmented salience (at the expense of valence) in the absence of interoceptive information (*Figure 5Eiii, iv*). This phenomenon is congruent with TE networks generated from data under conditions of CE$^{PKC\delta}$ inhibition, where CS-driven networks revert to uncertain/surprise states (*Figure 5C*). Our observations of enhanced decoding accuracy of exteroceptive stimuli in the network, along with a relative shift of feature importance towards primary sensory pIC (*Figure 2—figure supplement 2Aiii*), is congruent with the fundamentally different cognitive strategies ascribed to autism (*Happé and Frith, 2006*). These studies also show a dominance of posterior networks in perceptual tasks (*Koshino et al., 2005*). The shift towards pIC, which exhibits negative-valence bias (*Figure 1—figure supplement 5B*), may therefore explain augmented aversive behavior in these conditions.

Theories on affect, such as the somatic marker hypothesis (*Barrett, 2017*; *Bechara and Damasio, 2005*), suggest that interoceptive signals modulate decision-making and emotional learning. Generally, these theories propose that bodily states are integrated into affective decisions. Previous work highlighted IC–CE circuitry in controlling affective states (*Gehrlach et al., 2019*; *Schiff et al., 2018*; *Venniro et al., 2017*). In extension of these studies and our data, we propose that top-down information transfer in the IC↔CE/NBM network beteen IC and CE as a mechanism where interoceptive signals guide decision-making. Here, the magnitude of US responses along an antero-posterior valence gradient in the IC (*Figure 1—figure supplement 8B*) determines CS responses and conditioned responding at recall (*Figure 1—figure supplement 8A,C*). In this process, the IC not only represents sensory cues (*Livneh et al., 2017*), but also generates CS-associated allostatic states, instructing lower hierarchies to guide behavioral responding and memory formation (*Figure 2Ci*). Consistent with recent propositions (*Barrett and Simmons, 2015*), the gradual acquisition of CS information by the IC suggests the construction of a hierarchical task model in the interoceptive system that issues predictions about the physiological value of the CS to lower hierarchies. Thus, our study identifies a cortico-limbic hierarchy linking predictive representations of physiological states to decision making. Representations of CS and US synergize across IC–CE hierarchies for Pavlovian learning to optimize behavioral outcomes, potentially showcasing a general phenomenon in cortico-limbic interaction.

In conclusion, we propose that distributed neural ensembles in a cortico-limbic network ascribe affective value to sensory cues, and drive affective learning by recruiting interoceptive representations in the IC. Under states of value ambiguity, the CE drives bottom-up recruitment of the IC via the NBM. This, in turn, integrates stimuli with bodily states to potentially build interoceptive models in the IC, which then feed back to the CE to control adaptive behavioral decisions. In a psychiatric context, the inability to establish or recruit hierarchically organized interoceptive predictions in the IC↔CE/NBM circuitry based on the present sensory environment may contribute to symptoms of autism spectrum disorder or schizophrenia.

## Materials and methods

**Key resources table**

| Reagent type (species) or resource | Designation | Source or reference | Identifiers | Additional information |
|---|---|---|---|---|
| Strain, strain background (*M. musculus*, male) | wild-type | Charles River Laboratories | | C57BL/6J background |
| Strain, strain background (*M. musculus*, male) | PKCδ::Cre | doi:10.1038/nature09553 | Prkcd::GluClα::Cre | C57BL/6J background |

*Continued*

| Reagent type (species) or resource | Designation | Source or reference | Identifiers | Additional information |
|---|---|---|---|---|
| Strain, strain background (*M. musculus*, male) | SST::Cre | Jackson Laboratory | SOM-IRES::Cre; stock no: 013044 | C57BL/6J background |
| Other | DIO-GFP | This paper | AAV5.EF1a. DIO.GFP.WPRE | AAV vectors to transduce brain tissue; Titer:9.73E+10 |
| Other | syn-GFP | Penn Vector Core | AAV5.hsyn. eGFP.WPRE | AAV vectors to transduce brain tissue; Titer:1.15E+13 |
| Other | syn-ChR2 | Penn Vector Core | AAV5.hsyn. hChR2(H134R). eYFP.WPRE | AAV vectors to transduce brain tissue; Titer:1.87E+13 |
| Other | DIO-ChR2 | Penn Vector Core | AAV5.EF1a.DIO. hChR2(H134R). eYFP.WPRE | AAV vectors to transduce brain tissue; Titer:1.30E+13 |
| Other | syn-Arch | Penn Vector Core | AAV5.hsyn. ArchT.YFP.WPRE | AAV vectors to transduce brain tissue; Titer:4.68E+12 |
| Other | DIO-Arch | BI Biberach | AAV5. Ef1a.DIO. eArch.eYFP.WPRE | AAV vectors to transduce brain tissue; Titer:6.00E+12 |
| Other | DIO-NpHR | Penn Vector Core | AAV5.Ef1a.DIO. eNpHR3.0-eYFP.WPRE | AAV vectors to transduce brain tissue; Titer:2.59E+12 |
| Other | GCaMP6m | BI Biberach | AAV9.hsyn. GCaMP6m.WPRE | AAV vectors to transduce brain tissue; Titer:1.00E+12 |
| Other | DIO-GCaMP6f | Penn Vector Core | AAV1.hsyn.DIO. GCaMP6f.WPRE. | AAV vectors to transduce brain tissue; Titer:1.00E+13 |
| Other | AAV::Cre | Vector Biolabs | AAV5.CMV.Cre | AAV vectors to transduce brain tissue; Titer:1.00E+12 |
| Other | CAV::Cre | Montpellier Vector Platform | CAV2.Cre | CAV vectors to transduce brain tissue; Titer:5.50E+12 |
| Other | DIO-hM4 | Penn Vector Core | AAV5.hsyn.DIO. hM4D.mCherry. WPRE | AAV vectors to transduce brain tissue; Titer:1.01E+13 |
| Chemical compound, drug | DAPI | Life technologies | DAPI | 1 µg/mL |
| Chemical compound, drug | TZP | Sigma | CAS #147416-96-4 | (3 mg/kg) |

*Continued*

| Reagent type (species) or resource | Designation | Source or reference | Identifiers | Additional information |
|---|---|---|---|---|
| Antibody | Anti-PKCδ (mouse monoclonal) | BD Biosciences | Cat. #610398 | Lot#4080743 IF(1:1000) |
| Antibody | Anti-FOXO3/NeuN (chicken polyclonal) | Abcam | Cat. #ab131624 | Lot#GR88877-12 – IF(1:500) |
| Antibody | Anti-ChAT (goat polyclonal) | Millipore | Cat. #AB144P | Lot#2280814 – IF(1:200) |
| Antibody | Anti-mouse (goat polyclonal) | Life technologies | Cat. #A21052 | Lot#1712097 – IF(1:1000) |
| Antibody | Anti-chicken (goat polyclonal) | Life technologies | Cat. #A11041 | Lot#1383072 – (1:1000) |
| Antibody | Anti-goat (donkey polyclonal) | Abcam | Cat. #A11057 | Lot#819578 – IF(1:500) |
| Peptide, recombinant protein | Streptavidin-Alexa Fluor | Life technologies | Cat. #S11223 | Lot#18585036 – IF(1:1000) |
| Peptide, recombinant protein | CTB-Alexa Fluor | Invitrogen | Cat. #C34775 | |
| Software, algorithm | GraphPad Prism 7 and 8 | GraphPad Software, Inc | Version 8.1.1 | |
| Software, algorithm | scikit-learn package | doi:10.1007/s13398-014-0173-7.2 | Python 3 | |

## Animals

Male mice aged between 2 and 6 months were group housed in a colony on a 14 hr light/10 hr dark period and allowed water and food ad libitum, unless noted otherwise. Animal procedures were performed in accordance with institutional guidelines and were approved by the four respective Austrian (BGBl nr. 501/1988, idF BGBl I no. 162/2005) and European authorities (Directive 86/609/EEC of 24 November 1986, European Community) and covered by the license M58/002220/2011/9. Wild-type C57BL/6J mice were in-house bred and provided by the Research Institute of Molecular Pathology animal facility or ordered from Charles River Laboratories (strain C57BL/6J). Transgenic animals (Prkcd::GluClα::Cre [*Haubensak et al., 2010*] BAC transgenic mice, PKCδ::Cre and SOM-IRES::Cre transgenic mice, SST::Cre; stock no: 013044, Jackson Laboratory) were maintained on the C57BL/6J background. All mice were handled by the experimenters for several days prior starting any behavioral procedures.

## Resting state functional magnetic resonance imaging (resting state fMRI)

Animals ($CE_{sham}$/$CE_{NMDA}$) were subjected to resting state fMRI on a 15.2 T Bruker system (Bruker BioSpec, Ettlingen, Germany) with a 23 mm quadrature birdcage coil. Prior to imaging, all mice were anesthetized with 4% isoflurane, and care was taken to adjust the isoflurane levels immediately so that respiration did not fall below 140 breaths per minute (bpm) at any time. During imaging, respiration was maintained between 140 and 160 bpm. For the resting state fMRI study, a single shot echo planar imaging (EPI) sequence with spin echo readout was used (TR = 3000 ms, TE = 19.7 ms, FOV = $16 \times 16$ mm$^2$, voxel size = $250 \times 250$ μm$^2$, 30 slices 0.5 mm thick, one average, 240 repetitions, 12 min total imaging time). Following the resting state scan, a high-resolution T1-weighted anatomical scan was acquired using gradient echo sequence (TR = 500 ms, TE = 3 ms, FOV = $16 \times 16$ mm$^2$, voxel size = $125 \times 125$ μm$^2$, 30 slices 0.5 mm thick, four averages).

## Data processing for resting state fMRI

Resting state fMRI data were processed using the Data Processing Assistant for Resting-state fMRI Advanced Edition (DPARSF-A) toolbox, which is part of the Data Processing and Analysis of Brain Imaging (DPABI) toolbox version 2.1 (http://rfmri.org/dpabi) (*Chao-Gan, 2010*). The first 10 volumes were removed from each data set to ensure that steady state magnetization was reached. Data were processed in series of steps that included slice-timing correction, realignment, co-registration, normalization, and segmentation using in-house created mouse masks for cerebrospinal fluid (CSF), white matter (WM), and gray matter (GM). Nuisance covariates related to motion were regressed out using Friston 24-parameter model (*Friston et al., 1996*). In addition, WM and CSF mean time-series were used as nuisance regressors in the general linear model to reduce influence of physiological noise (*Margulies et al., 2007*). Data were analyzed with and without linear regression of global signal (*Murphy et al., 2009*; *Murphy and Fox, 2017*; *Saad et al., 2012*). Data were spatially smoothed with a 2.4 pixel full-width half-maximum Gaussian kernel. A narrow band pass filter (0.054-0.083 Hz) (*Wee et al., 2012*) was used following nuisance regression. All data were co-registered to the in-house generated mouse atlas with 80 distinct brain regions. For the seed-based functional connectivity analysis, the mean time series signal from the region of interest (seed) was calculated and correlated with the time series signal from each pixel of the brain. Between group comparison was done using pairwise t-test followed by Gaussian Random Field (GRF) Theory Multiple Comparison Correction (voxel-level p-value=0.05, cluster-level p-value=0.05). Within group comparison was done using one-sample t-test followed by GRF multiple comparison correction (voxel-level p-value=0.05, cluster-level p-value=0.05). For the functional connectivity matrix, mean time course signal from 80 brain region was calculated. Fisher's z-transformed Pearson correlation coefficients between each pair of brain regions were calculated for all groups (*Song et al., 2011*). One-sample t-test was used to find a significant pair of brain regions within a group, with $p<0.05$ considered significant. All analyses were performed using freely available R-project software (*R Development Core Team, 2011*). The network visualization was performed with BrainNet Viewer (*Xia et al., 2013*). Resting state fMRI results shown here use global signal regression (GSR). An alternative approach for noise correction was also performed (*Behzadi et al., 2007*), and no significant differences among results were found (data not shown). We chose to interpret results following GSR, as this approach improved specificity of positive correlations (*Fox et al., 2009*; *Weissenbacher et al., 2009*) and aided in symptom prediction following focal brain lesions in humans (*Boes et al., 2015*).

## Stereotactic surgery for virus/toxin injection, fiber-optic cannula/lens/electrode implantations

General surgical procedures: Mice were deeply anesthetized with isoflurane and maintained at 1.5-2% throughout the procedure (Univentor 400). Animals were mounted in a stereotactic frame (Kopf), while body temperature was kept constant at 36°C via a rectal temperature-controlled heating pad (FHC). Before incision, local anesthesia was provided underneath the skin by injection of 0.1 ml of lidocaine (Xylanaest, 1%). The exposed skull was drilled through above the area of interest, relative to bregma (*Paxinos and Franklin, 2007*). Animals were provided with post-surgical analgesics (250 mg/ml Carprofen; Rimadyl, Pfizer) and antibiotics (400 mg/l Enrofloxacin; Baytril, KVP pharma) via drinking water for 7 days. *Opto-chemogenetic experiments:* For optogenetic experiments, animals were bilaterally injected with the appropriate viruses (CEl 80 nl, AP -1.38, ML ±2.9, DV -4.85 mm; aIC 100 nl, AP +1.54, ML ±3.17, DV -3.55; pIC 80 nl, AP -0.7, ML ±4.2, DV -4 mm) and bilaterally implanted with fiber-optic cannulas (Doric Lenses, 200-400µm, NA 0.37-0.53) 0.5 mm above the target coordinates (CEl AP -1.38, ML ±2.9, DV -4.35 mm; NBM AP -0.4, ML ± 1.6, V -4.3 mm). For chemogenetic inhibition, we used the Cre-dependent hM4 DREADD system (AAV::DIO-hM4) injected bilaterally into aIC (100 nl) or pIC (80 nl). The Cre-expressing construct (CAV::Cre) was delivered bilaterally to pIC or CEl. For $Ca^{2+}$ imaging, mice were unilaterally injected with an AAV carrying a $Ca^{2+}$ indicator into CEm (60 nl, AAV::GCaMP6m; AP -1.06, ML +2.25, DV -4.5 mm) or into CEl (50 nl, AAV::DIO-GCaMP6f; AP -1.38, ML +2.9, DV -4.85 mm). At $\geq$4 weeks post-injection, a lens was implanted above the injection site (Inscopix microendoscope 0561 Part ID:1050-002182). After a 1-week recovery period, the baseplate was cemented onto the skull (Inscopix microscope baseplate V2, Part ID:1050-002192). For in vivo electrophysiology, silicon probes

(single-site; Neuronexus) or custom-built tetrodes (multi-site; 30 µm Nichrome wires, California Fine Wire; two bundles per site) were affixed to fiber-optic cannulas and implanted. Ground screws were mounted above the contralateral prefrontal cortex and cerebellum. All implants were fixed to the skull with dental cement (SuperBond C&B kit, Prestige Dental Products).

## In vivo electrophysiology and data acquisition

Mice were handled and habituated to the recording room for several days prior to experimental recordings. Implanted electrodes were connected, via an Omnetics connector, to a 16-channel unity-gain headstage (Plexon), after which mice were left in the home cage for 10 min. The headstage was connected to a pre-amplifier, and the signal was band-pass filtered (3 Hz-1khz) and amplified. Neural activity was digitized at 40 kHz and highpass-filtered for spikes (800 Hz) and LFPs (3–200 Hz) for off-line analysis. Spikes were sorted with Offline Sorter v4 (OFS, Plexon). All recording sessions for each mouse were merged, and principal component (PC) analysis was performed on unsorted waveforms. Spikes were manually sorted with OFS. Single units were sorted manually in 3D PC feature space for each session and declared a single unit if the spike cluster was separable from noise and other clusters and no refractory period infringements were detected. To avoid multi-sampling of single units, cross-correlograms of units from adjacent channels were inspected for co-firing and respective units removed from analysis.

## Ca$^{2+}$ imaging and data acquisition

Deep-brain calcium imaging was performed with an in vivo miniature endoscope (Inscopix). Mice were handled and habituated to the mounted microscope for several days prior to experimental recordings. nVista HD System v2.0.32 (Inscopix) was used for the acquisition of Ca$^{2+}$ signals. Images were obtained at 20 fps with automatically set exposure time, 3.25 gain, and LED power set to 40%. Data was processed and analyzed with Mosaic v1.2.0 software (Inscopix). The aligned videos were down-sampled 2x2 (time x space) and the Ca$^{2+}$ signal was calculated as the relative change of fluorescence over the entire recording session ($\Delta F(t)/F0=(F(t)-F0)/F0$). The individual neurons and their Ca$^{2+}$ traces were extracted by applying PCA-ICA analysis. Spatial filters obtained by PCA-ICA were then manually selected to avoid duplicates or false units in further analysis. Ca$^{2+}$ traces were then filtered (0.5 Hz low pass filter) and automated Ca$^{2+}$ event detection was applied ($\Delta F(t)/F0 > 3xMAD$ (median absolute deviation), $\tau_{off}=0.2$ s). Exported events were further analyzed with Neuroexplorer software v5.114 (Plexon).

## Peri-event time histogram (PETH) analysis of neural recordings

Data from in vivo electrophysiology and calcium recordings were processed in Neuroexplorer. Neuronal firing and calcium signals were extracted as 500 ms binned events. Neuronal events were then exported as PETH and z-scored per recording stage. Only data within -8 – 18s relative to CS onset was considered and smoothed with a Gaussian filter (degree of 5 for IC and 8 for CE data). The electrical shock artefact was masked, and neural activity originating from a channel showing prolonged LFP black-out at a given trial was replaced with the population average of the same bin.

## Behavioral design for in vivo electrophysiological experiments

Mice underwent 3 habituation sessions (6 presentations per CS in blocks of 2) and 3 port training sessions (random water delivery at the port), each 30 min after intraperitoneal injection of either PBS, CNO, or TZP (treatment order counterbalanced). For RC, mice were separated into a PBS and CNO groups, receiving respective daily intraperitoneal injections. After 8-12 RC sessions (20 CS-US pairings/session), mice were subjected to an FC session (3-4 CS-US pairings), receiving the same treatment as in RC. After three to four recall sessions (using the same treatments as in habituation, four to six presentations per CS in blocks of two), mice underwent single RC and FC sessions with the respective converse treatment (PBS or CNO), followed by three recall sessions, each with a different treatment (PBS, CNO, or TZP). Reward-specific behavior was scored when a mouse broke the IR beam while entering the port ('port visits'), whereas freezing onsets were scored (1s minimum time immobile, 1s sliding window, Motion Threshold=80) on recorded videos with Cineplex Editor v3.6 (Plexon) and aligned to electrophysiological data offline.

## Behavioral design for Ca²⁺ imaging experiments

Mice underwent two habituation sessions with four presentations of each CS in blocks of two and two port training sessions (random water delivery in the port). Thirty minutes before each session, mice received an intraperitoneal injection of either PBS or CNO (treatment order was counterbalanced). All mice subsequently underwent 6-10 RC sessions with 12 CS-US pairings, receiving a daily intraperitoneal injection of PBS before RC sessions, and one session with a prior CNO injection. Next, mice were subjected to two FC sessions with two CS-US pairings each, receiving an injection of either PBS or CNO (in balanced order). Thereafter, all mice were subjected to 4 recall sessions (two PBS and two CNO sessions). Reward-specific behavior was scored when a mouse broke the IR beam while entering the port ('port visits'), whereas freezing onsets were scored on recorded videos with Ethovision v12.0 (Noldus) offline (1s minimum time immobile, <0.5% area change for a 1s sliding window).

## Neural decoding

Neural decoding was performed on raw recorded neural data (X) to determine the representation of stimuli (y) within the recorded brain regions. We reasoned that operations on raw data, while not maximizing decoder accuracy, will allow for more straightforward comparisons between conditions, as minimal non-linearities introduced by independent data pre-processing steps are minimized. Decoding was performed by solving classification problems (y=f(X)) with classes y (defined for Task 1 'CS': bins before CS onset, bins during CS; for Task 2 'US': bins before CS, bins after US).

Three different types computation were performed: 1. Single-region decoding, 2. identification of similarity between neural activity patterns for single regions, and 3. multi-region decoding. The computations were performed using Jupyter Notebooks, Python 3, and the scikit-learn package (*Fabian et al., 2011*). 1s bin data was used for all the computations.

1. Single-region decoding. The neural data matrix (X) was combined from all mice and defined by region: per stage, treatment, CS, and day. The alignment was performed based on the classification goal y. Before classification, the data was z-scored and balanced by under-sampling. The Multi-layer Perceptron classifier was used. A 5-fold cross validation was performed, and the procedure was repeated 40 times. The mean accuracy of all iterations was used as the criterion for decoder performance. The best single neurons in CE were defined as those reaching highest accuracy when X consisted of a single neuron only (see *Figure 5—source data 1* for all neurons). For region-wise decoding, neuron selection versions were applied according to the maximum number of neurons available to allow meaningful comparisons between treatments and stages, as indicated in the respective figure legends. As a control, the classification procedure was applied to shuffled class vectors y for each task.

2. Similarity of neural activity. To evaluate the similarity of the representations of conditioned and unconditioned stimuli within neuronal activity over time, decoders trained on one stimulus were applied to another stimulus within the same stage. Four combinations were performed: (1) lick on R-CS, (2) shock on F-CS, (3) R-CS on F-CS and (4) F-CS on R-CS. For each combination, a decoder was trained 10 times on one stimulus and applied on the second one. As a control, The trained classifier was applied to shuffled target class vectors y.

3. Discrimination of neural activity. To evaluate the ability to discriminate between the two CSs, three classes were defined: class 0 (bins before the CSs), class 1 (R-CS bins), and class 2 (F-CS bins). The same criteria used for single-region decoding were applied to the selection of random/best neurons and training/evaluation of the classification. Evaluation consisted of two steps: (1) classical accuracy considering all three classes (data not shown) and (2) a sub-selection of (1) with class 0 omitted. This resulted in the accuracy of assigning CS bins to the correct CS divided by the number of all CS bins, which were also classified as CS bins.

4. Multi-region decoding. All available neural data from all mice and regions were combined into data matrices (X) as 'network' and defined: per stimulus (CS or US) and stage. alignment was performed as for 'Single-region decoding' based on the classification goal. Two different treatments were investigated: (i) Control: only data from control sessions for all regions (PBS), (ii) hM4(CE$^{PKC\delta}$): only data from CNO sessions for regions aIC and pIC and PBS sessions for CE$^{PKC\delta}$, CE$^{SST}$, and CEm. Prior to Random Forest classification, the data were z-scored and balanced by under-sampling. 100 neurons were selected randomly, although the percentage distribution between the regions was respected. A 5-fold cross validation was performed, and the procedure was repeated 40 times. In addition to the mean classification accuracy of all iterations, the mean feature importance of all single neurons for each region was computed.

## Combined Pavlovian reward and fear conditioning for behavioral cohorts

Animals from all experimental cohorts were water deprived for 16 hr at all stages of the experiment, while their weight was continuously monitored to ensure it never fell below 80% of their initial weight. Prior to conditioning experiments, animals underwent a port training session where they learned to associate the port with the delivery of a water drop in context A (light on, water delivery port, neutral grid). Only after successful port training did the animals proceed to reward conditioning (RC). All cohorts underwent at least 8 RC sessions in context A, where they received between 12 and 24 pairings of a neutral sound (50 ms white noise, 0.9 Hz for 10 s at 70dB, 'R-CS') with the subsequent delivery of a water drop (valve opened for 1s). Thereafter, mice underwent a single fear conditioning (FC) session in context B (no light, port removed, shock grid) where they received five pairings of a different neutral sound (3kHz continuous for 10s at 70dB, 'F-CS') with the delivery of a mild 1s foot shock (0.5 mA, Coulbourn). Memory testing was conducted in context A by presenting both unreinforced sounds four times each interleaved in blocks of two (2x(2R-CS + 2F-CS)). Reward-specific behavior was scored when a mouse broke the IR beam while entering the port ('port visits'), whereas freezing behavior was scored on recorded videos with Ethovision v12.0 (Noldus) offline (1s minimum time immobile, <0.5% area change for a 1s sliding window).

## Circuit manipulations

For optogenetic manipulations, mice were handled and habituated to attachment of the fiber-optic patch cord (Doric Lenses) to the fiber implants for several days prior to the experiment. For behavioral cohorts, activation of Channelrhodopsin-2 (ChR2) was achieved with a 473 nm laser, delivering 10 ms pulses at an intensity of 10 mW at the fiber tip at a stimulation frequency of 20 Hz for IC projections to CE. Neuronal inhibition was achieved by activation of Halorhodopsin or Archaerhodopsin using an 489 nm laser at constant 7-8 mW light intensity at the fiber tip. Intensity was adjusted before experiments with a power meter (Thorlabs, PM100D). The laser was triggered by a custom Matlab (v2014b) script during conditioning experiments for conditioned stimulus (CS) periods only. CE–NBM stimulation during in vivo electrophysiological recordings was performed with 5 ms pulses from a 470 nm LED (Doric Lenses). For chemogenetic/pharmacological manipulations, mice were handled and habituated to intraperitoneal PBS injections for 3 days. PBS, CNO (Sigma), and TZP (Sigma) injections were performed 30 min prior to the start of the experiment, and mice were returned to their home cage after injection. Volume was adjusted to 0.1 ml for all experiments. A final dosage of 3 mg/kg for TZP and 5 mg/kg for CNO was used for all chemogenetic experiments other than RC sessions, for which the dosage was adjusted to 2.5 mg/kg.

## Transfer entropy

Transfer entropy $\mathbf{TE_{n_1-n_2}}$ between neurons $n_1$ and $n_2$ was computed using the Python package PyInform (https://github.com/ELIFE-ASU/PyInform), which is a wrapper of the inform library using Jupyter Notebooks and Python 3. For each treatment, a sound and stage 'network' (as for the multiregion decoding) was created with 1s bin data. 500 neurons were subsequently drawn randomly from this matrix, considering the percentage distribution between the regions. The TE was computed pairwise between all neurons. The local maximum per pair was taken. Only the upper 50% of all pairs per region combination were considered. TE between regions was defined by the average TE of neurons belonging to the regions (as in *Lizier et al., 2011*).

$$TE_{k,v}(n_1 - n_2) = \left\langle TE_k(n_{1,i} - n_{2,j})_{i,j} \right\rangle$$

Where $k$ refers to past states and $i$ and $j$ label the sample subset of $\text{Region}_{a,i}$ and $\text{Region}_{b,j}$ of size $v$ in each region.

Significance was tested as in *Timme and Lapish, 2018*. The null hypothesis was that $n_2$ does not depend on $n_1$. 1000 surrogate datasets were created by shuffling the time-series and computing the region-wise TE. The proportion of $TE_{surrogate} >= TE_{real}$ was used as the p-value for significance testing ($\alpha < 0.05$).

## Brain slice preparation and electrophysiology

Three weeks prior to electrophysiological recordings, male WT mice received injections of AAV-ChR2 in the IC, while transgenic SST- and PKCδ::Cre mice received injections of AAV-DIO-ChR2 in the CE. At 2–3 months of age, mice were deeply anesthetized with isoflurane, decapitated, and their brains quickly chilled in sucrose-based dissection buffer bubbled with 95% $O_2$/5% $CO_2$ containing the following (in mM): 220 Sucrose, 26 $NaHCO_3$, 2.4 KCl, 10 $MgSO_4$, 0.5 $CaCl_2$, 3 Sodium Pyruvate, 5 Sodium Ascorbate, and 10 glucose. Coronal brain slices (300 µm thick) were cut in dissection buffer using a Vibratome (Leica, VT1000S), and immediately incubated for a 15 min recovery phase in oxygenated artificial cerebrospinal fluid (aCSF) comprised of the following (in mM): 126 NaCl, 2.5 KCl, 1.25 $NaH_2PO_4$, 26 $NaHCO_3$, 2.5 $CaCl_2$, 2.5 $MgCl_2$, and 25 glucose in 95% $O_2$/5% $CO_2$ at 32℃. This was followed by a slice resting phase with oxygenated aCSF for at least 45 min at room temperature (RT). Individual brain slices containing target regions (CE for IC injections, NBM for CE injections) were placed on the stage of an upright, infrared-differential interference contrast microscope (Olympus BX50WI) mounted on a X-Y table (Olympus) and visualized with a 40x water immersion objective by an infrared sensitive digital camera (Hamamatsu, ORCA-03). Slices were fully submerged and continuously perfused at a rate of 1–2 ml per min with oxygenated aCSF. Patch pipettes were pulled on a Flaming/Brown micropipette puller (Sutter, P-97) from borosilicate glass (1.5 mm outer and 0.86 mm inner diameter, Sutter) to final resistances ranging from 3 to 5 MΩ. The Internal solution for recording responses to optogenetic stimulation of PKC-δ/SST neuronal input to NBM contained the following (in mM): 135 KCl, 0.2 EGTA, 10 HEPES, 2 MgATP, 0.5 $Na_2$GTP, 10 $Na_2$phosphocreatine, and 0.2% (w/w) Biocytin. For recording responses to optogenetic stimulation of IC neuronal input in CE, the internal solution contained the following (in mM): 135 K-Gluconate, 5 KCl, 10 HEPES, 2 $MgCl_2$, 0.2 EGTA, 1 $Na_2$ATP, 0.4 NaGTP, 10 $Na_2$Phosphocreatine, 0.2% (w/w) Biocytin, and 280–290 mOsmol. Membrane currents were recorded with a Multiclamp 700B amplifier (Molecular Devices). Electrophysiological signals were low-pass filtered at 3 kHz, sampled at 10 kHz (Digidata 1440A, Axon Instruments) and further analyzed with pClamp 10 software (Molecular Devices). Recordings started 5 min after letting the cell reestablish constant activity post break-in. Inputs from IC to CE or CE to NBM were stimulated in voltage-clamp (−70 mV) with 20 ms blue light pulses through a 40x electrophysiology microscope objective, driven by a 120W mercury lamp (X-Cite 120 PC Q). The amplitude of 4 pulses, 1 s apart, was averaged as postsynaptic responses of specific cell types in the CE or NBM. Cell identity was confirmed using biocytin and post hoc immunohistochemistry.

## Histological evaluation

For verification of injection targeting, implant placement, and virus expression, mice were deeply anesthetized by an intraperitoneal injection of a mixture of Ketamine (10 mg/ml, OGRIS Pharma) and Medetomidine (Domitor, ORION Pharma) in phosphate-buffered saline (PBS), and transcardially perfused with cold 10 ml PBS and 30 ml of 4% Paraformaldehyde (PFA). Brains were immediately removed and post-fixed overnight in 4% PFA at 4℃. 20µm cryo-sections were obtained from brains from all cohorts except animals subjected to electrophysiological recordings or $Ca^{2+}$ imaging, for which 80-µm-thick vibratome sections were collected.

## Immunohistochemistry

Sections were permeabilized with PBS-T (0.1% Triton X-100 in PBS or 0.2% for ex vivo electrophysiology sections) and subsequently blocked with 2% bovine serum albumin (BSA, in PBS-T) for 1 hr to attenuate unspecific binding. Slides were incubated overnight with primary antibodies (Key Resources Table) in BSA at 4℃. Slides were then washed in PBS-T and incubated with fluorescently conjugated secondary antibodies (Key Resources Table) in BSA for 2h at room temperature. After washing, slides were mounted with fluorescence mounting medium (Dako) and images were acquired on a confocal microscope (Zeiss) and slide scanner (3DHistech).

## Data analyses and statistical tests

Sample sizes were in line with estimates derived from previous experiments using G*Power Version 3.1.9.6. For neural recording experiments, three to five animals were required (effect size 0.3; *Groessl et al., 2018*). For behavioral experiments, the target sample size was in the range of 8-10

animals (effect size 0.45, *Groessl et al., 2018*). Animals were randomly assigned to experimental cohorts. The behavioral experimenter was blind to the treatment wherever possible. Behavioral and neural data analyses was carried out blinded and/or computationally wherever applicable. Establishment of the behavioral assay, neural recordings, and circuit manipulation were performed in independent experiments with separate animal cohorts (*Figures 2*, *5*, *Figure 2—figure supplements 5*, *7*, *Figure 5—figure supplement 3*; biological replicates). Basic behavior was replicated across experiments for control groups. Circuit manipulations were replicated using different technologies on separate experiments and cohorts (*Figures 2*, *Figure 2—figure supplements 5*, *7*; biological replicates). Neural activity recordings were replicated in independent animals (biological replicates) and across sessions within animals (technical replicates) (*Figures 1*, *5*, *Figure 1—figure supplements 5*, *6* and *8*, *Figure 5—figure supplement 5*). For behavioral experiments, 8/97 animals were excluded for failing port training, low virus expression, or misplaced/broken fibers. For in vivo electrophysiology and calcium imaging, 4/14 and 13/26 animals were excluded due to absent Calcium signals or absent/low quality signals, respectively. After unit identification, no further animals were excluded in either case. Statistical significance was determined using parametric statistics (assuming normality of the data) or permutation tests. All statistical tests were performed using Graph Pad Prism (versions 7 & 8) and custom R and/or Python codes. Significant results are indicated as described in the figure legends and *Appendix 1—table 1*.

## Acknowledgements

We thank Manuel Pasieka from Bioinformatics and Scientific Computing, Vienna Biocenter (VBC) for development of software used in behavioral experiments video recording and custom Matlab code for behavioral and neuronal data analysis. We thank Nadia Kaouane for establishing the reward/fear Pavlovian conditioning paradigm. We also thank Lydia Zopf and Peter Opriessnig from Preclinical Imaging, Vienna BioCenter Core Facilities (pcIMAG, VBCF) for fMRI data acquisition and analysis. We thank Tamara Engelmaier and Mihaela Zeba from Histopathology (VBCF) for tissue processing. Additionally, we want to acknowledge Thomas Lendl from BioOptics (IMP) for generating custom Java scripts for histological image analysis and Maria Novatchkova from BioInformatics (IMP) for statistical analysis. We thank Life Science Editors for editing assistance. This research was supported by the Research Institute of Molecular Pathology (IMP), Boehringer Ingelheim, the Austrian Research Promotion Agency (FFG), and a grant from the European Community's Seventh Framework Programme (FP/2007–2013)/ERC grant agreement no. 311701. We thank Life Science Editors for editorial assistance. VRVis is funded by BMK, BMDW, Styria, SFG and Vienna Business Agency in the scope of COMET - Competence Centers for Excellent Technologies (854174) which is managed by FFG.

## Additional information

### Funding

| Funder | Grant reference number | Author |
|---|---|---|
| European Commission | 311701 | Wulf Haubensak |
| Austrian Research Promotion Agency | 852936 | Wulf Haubensak |
| Institute of Molecular Pathology (IMP) | | Wulf Haubensak |
| Boehringer Ingelheim | | Wulf Haubensak |
| Austrian Research Promotion Agency | 854174 | Katja Buehler |

The funders had no role in study design, data collection and interpretation, or the decision to submit the work for publication.

## Author contributions
Dominic Kargl, Joanna Kaczanowska, Conceptualization, Data curation, Formal analysis, Investigation, Visualization, Methodology, Writing - original draft, Project administration, Writing - review and editing; Sophia Ulonska, Data curation, Software, Formal analysis, Methodology, Writing - review and editing; Florian Groessl, Data curation, Formal analysis, Investigation, Visualization, Writing - review and editing; Lukasz Piszczek, Data curation, Software, Formal analysis, Writing - review and editing; Jelena Lazovic, Data curation, Formal analysis, Investigation, Visualization, Methodology, Writing - review and editing; Katja Buehler, Supervision, Funding acquisition, Validation, Methodology, Writing - review and editing; Wulf Haubensak, Conceptualization, Supervision, Funding acquisition, Validation, Writing - original draft, Project administration, Writing - review and editing

## Author ORCIDs
Dominic Kargl 
Joanna Kaczanowska 
Lukasz Piszczek 
Katja Buehler 
Wulf Haubensak 

## Ethics
Animal experimentation: Animal procedures were performed in accordance with institutional guidelines and were approved by the 4 respective Austrian (BGBl nr. 501/1988, idF BGBl I no. 162/2005) and European authorities (Directive 86/609/EEC of 24 November 1986, European Community) and covered by the license M58/002220/2011/9.

## Decision letter and Author response
Decision letter https://doi.org/10.7554/eLife.60336.sa1
Author response https://doi.org/10.7554/eLife.60336.sa2

# Additional files
## Supplementary files
• Transparent reporting form

## Data availability
All data generated or analyzed during this study are included in the manuscript and supporting files (Figure 1-source data 1, Figure 1-figure supplement 1-source data 1, Figure 2-source data 1, Figure 3-source data 1, Figure 4-source data 1, Figure 4-figure supplement 1-source data 1, Figure 5-source data 1, Figure 5-source data 2).

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

## Appendix 1

### Neuronal responses to task stimuli

The Fraction of responders in the IC/CE was based on a trial-averaged z-score deflection of ±1.65. Population responses to task stimuli are shown in *Figure 1—figure supplements 5A–C* and *6A–C*. PETH of positive CS responders in the IC/CE are shown in *Figure 1—figure supplements 5Aiii, Ciii* and *6Aiii,Ciii*. Single neuron activity suppression was virtually absent with these criteria, therefore suppression is only reflected on the population level. Consistent conditioned responding reflects task performance. Since performance at recall varied within the IC/CE groups (*Figure 1—figure supplement 8A*), we segregated neurons of individual mice into 'performers' and 'non-performers' using a median split on behavioral performance at recall. For IC subregions, we found a striking difference in US response magnitude between the two groups, where non-performers population response to R-US in aIC and tonic response to F-US in pIC were largely absent. These data indicate that successful US encoding in IC determines behavioral performance (*Figure 1—figure supplement 8Bi, iv*). This pattern transferred to the CS in performers, where aIC showed population responses to R-CS, whereas pIC displayed responses to F-CS at recall (*Figure 1—figure supplement 8Ci, iv*). Similarly, a median split based on behavioral performance at recall for the CE recording cohort (*Figure 1—figure supplement 8A*) revealed significant differences in US response magnitude in the PKCδ$^+$ population for the fear domain, and a trend for the reward domain upon reward delivery (*Figure 1—figure supplement 8Dv, iv*). This suggests that PKCδ$^+$ neurons may signal the need to learn, while their level of engagement may determine success.

### IC–CE circuit architecture

To address whether IC-CE TE for conditioned responding (*Figure 2C*) may emerge from an underlying neural network architecture, we performed retrograde anatomical tracing with fluorescently labeled CTB. We injected CTB into the CE and quantified CTB$^+$ neurons in the IC relative to DAPI of the entire IC area and the respective projection field to the CE. We found that CE-projecting neurons are more abundant in pIC than aIC relative to its size or projection area (*Figure 2—figure supplement 1A*). TE for conditioned responding could suggest a biased innervation of CE subpopulations by IC subregions. We examined whether TE maps are reflected in the circuit architecture by assaying synaptic connectivity using slice electrophysiology combined with optogenetics. PKCδ::Cre mice received an injection of AAVs carrying Cre-dependent GFP into the CE to allow for direct identification of SST$^+$ (approximated by absence of GFP expression) and PKCδ$^+$ neurons, and syn-ChR2 was injected into aIC or pIC for pan-neuronal expression (*Figure 2—figure supplement 1B*). Optogenetic excitation of a/pIC input to CE revealed monosynaptic connections between IC and CEl neurons. Interestingly, we found no difference in the synaptic innervation between CEl populations (*Figure 2A*, 92% of PKCδ$^+$ and 91% of SST$^+$ neurons responded to aIC/100% of PKCδ$^+$/SST$^+$ neurons to pIC input). These data support a functional rostro-caudal organization of the IC-CE network, reflecting differential US tuning in IC subregions (*Figure 1—figure supplement 5B*). Based on these data, we propose that functional differences in subnetworks emerge from distributed ensembles rather than from a pre-determined circuit architecture.

### CE–NBM circuit architecture

We assessed the anatomical connectivity between CE and NBM by injecting the retrograde tracer cholera toxin-B (CTB) into the NBM, which showed robust backlabeling in CE. Double-staining for PKCδ revealed that this projection is dominated by the PKCδ$^+$ population (~10% of CTB$^+$/DAPI are PKCδ$^+$ vs. ~5% PKCδ$^-$) (*Figure 4—figure supplement 2A*). Backlabeling to CEm was previously reported to be sparse or absent (*Jolkkonen et al., 2002*). Since the vast majority of CE neurons are GABAergic (*Cassell et al., 1999*), we suspected that a disinhibitory mechanism may gate (cholinergic) output neurons in the NBM. To examine cell type-specific innervation of NBM by CE, we performed slice electrophysiology, combined with optogenetic stimulation of CE fibers in NBM (*Figure 4—figure supplement 2Bi*). An AAV carrying Cre-dependent Channelrhodopsin-2 (DIO-ChR2) was injected into the CE of PKCδ::Cre and SST::Cre mice, and slices containing NBM were obtained after 3 weeks. Patch-clamp was guided by cell morphology, as corticopetal neurons are

magnocellular (mc), rather than parvocellular (pc) interneurons (IN) (*Gritti et al., 1997*). To identify cholinergic cells, neurons were filled with biocytin for labelling with fluorescently tagged streptavidin, and stained for choline acetyltransferase (ChAT) post hoc (*Figure 4—figure supplement 2Bii*). Of recovered mc neurons, 33% (2/6) were identified as ChAT$^+$ neurons, but no pc neurons stained for ChAT (0/14). Optogenetic stimulation of CE$^{SST}$ and CE$^{PKC\delta}$ neuronal inputs induced inhibitory postsynaptic responses in 82% (9/11) and 77% (10/13) of ChAT$^-$ IN, respectively. We found that 14% (1/7) of mc neurons are responsive to CE$^{SST}$ but none responded to CE$^{PKC\delta}$ input (0/7; *Figure 4C*), consistent with previous reports showing that CE axons largely avoid ChAT$^+$ neurons (*Jolkkonen et al., 2002*).

**Appendix 1—table 1.** Detailed statistical report for MANOVA/ANOVA analyses.

| Dataset | Statistical test | p-values | F ratio (DFn, DFd) |
|---|---|---|---|
| *Figure 1Ci* CS IC<br>Stage x Subregion<br>Stage<br>Subregion | Two-way ANOVA | <br>p<0.0001<br>p<0.0001<br>p<0.0001 | <br>$F_{9,6384}=13.69$<br>$F_{3,6384}=29.11$<br>$F_{3,6384}=203.31$ |
| *Figure 1Cii* CS similarity IC<br>Stage x Subregion<br>Stage<br>Subregion | Two-way ANOVA | <br>p<0.0001<br>p<0.0001<br>p<0.0001 | <br>$F_{3,6392}=42.10$<br>$F_{1,6392}=116.62$<br>$F_{3,6392}=42.69$ |
| *Figure 1Cii* CS←US transfer IC<br>Stage x Subregion<br>Stage<br>Subregion | Two-way ANOVA | <br>p<0.0001<br>p<0.0001<br>p<0.0001 | <br>$F_{3,6392}=50.14$<br>$F_{1,6392}=102.10$<br>$F_{3,6392}=469.13$ |
| *Figure 1Di* CS CE<br>Stage x Population<br>Stage<br>Population | Two-way ANOVA | <br>p<0.0001<br>p<0.0001<br>p<0.0001 | <br>$F_{15,9576}=9.30$<br>$F_{3,9576}=30.92$<br>$F_{5,9576}=28.20$ |
| *Figure 1Dii*<br>CS similarity CE<br>Stage x Population<br>Stage<br>Population | Two-way ANOVA | <br><br>p<0.0001<br>p<0.0001<br>p<0.0001 | <br><br>$F_{5,9588}=30.40$<br>$F_{1,9588}=25.90$<br>$F_{5,9588}=33.87$ |
| *Figure 1Diii*<br>CS←US transfer CE<br>Stage x Population<br>Stage<br>Population | Two-way ANOVA | <br><br>p<0.0001<br>p<0.0001<br>p<0.0001 | <br><br>$F_{5,9588}=339.60$<br>$F_{1,9588}=88.70$<br>$F_{5,9588}=798.90$ |
| *Figure 2Dii* Conditioning early | MANOVA | p=0.0126 | $F_{2,44}=3.60$ |
| *Figure 2Dii* Conditioning late | MANOVA | p=0.0004 | $F_{2,44}=6.43$ |
| *Figure 2Diii*, left Recall opto | MANOVA | p=0.0050 | $F_{1,13}=8.18$ |
| *Figure 2Diii*, right Recall hM4(pIC) | MANOVA | p=0.0067 | $F_{1,17}=6.81$ |
| *Figure 3Cii* Recall hM4(aIC)-pIC | MANOVA | p=0.0471 | $F_{1,18}=3.64$ |
| *Figure 4Dii* CE$^{PKC\delta}$-NBM stim.<br>aIC pIC | One-way RM ANOVA | p<0.0001<br>p<0.0001 | $F_{1,116,11,16}=153.00$<br>$F_{1,340,14,74}=23.60$ |
| *Figure 4Diii* CE$^{PKC\delta}$-NBM stim. | One-way ANOVA | p<0.0001 | $F_{2,32}=209.40$ |
| *Figure 5Dii* Recall CE$^{PKC\delta}$-NBM | MANOVA | p=0.0045 | $F_{1,10}=9.76$ |
| *Figure 5Eii* CS CE<br>Treatment x Population<br>Treatment<br>Population | Two-way ANOVA | <br>p<0.0001<br>p<0.0001<br>p<0.0001 | <br>$F_{5,4788}=117.50$<br>$F_{1,4788}=102.80$<br>$F_{5,4788}=449.30$ |
| *Figure 5Eiii* CS similarity CE<br>Treatment x Population<br>Treatment<br>Population | Two-way ANOVA | <br>p<0.0001<br>p<0.0001<br>p<0.0001 | <br>$F_{5,9588}=306.50$<br>$F_{1,9588}=134.80$<br>$F_{5,9588}=340.70$ |

*Appendix 1—table 1 continued*

| Dataset | Statistical test | p-values | F ratio (DFn, DFd) |
|---|---|---|---|
| *Figure 5Eiv* CS←US transfer CE | Two-way ANOVA | | |
| Treatment x Population | | p<0.0001 | $F_{5,9588}$=163.90 |
| Treatment | | p<0.0001 | $F_{1,9588}$=432.50 |
| Population | | p<0.0001 | $F_{5,9588}$=584.50 |
| *Figure 1—figure supplement 5Ai* Habituation R-CS | Two-way RM ANOVA | | |
| Time x Subregion | | p=0.1764 | $F_{52,11180}$=1.18 |
| Time | | p=0.0037 | $F_{52,11180}$=1.61 |
| Subregion | | p=0.7836 | $F_{1,215}$=0.08 |
| *Figure 1—figure supplement 5Ai* Habituation F-CS | Two-way RM ANOVA | | |
| Time x Subregion | | p<0.0001 | $F_{52,11076}$=2.23 |
| Time | | p<0.0001 | $F_{52,11076}$=2.17 |
| Subregion | | p=0.9293 | $F_{1,213}$=0.008 |
| *Figure 1—figure supplement 5Aiii* Habituation R-CS Responders | Two-way RM ANOVA | | |
| Time x Subregion | | p=0.8104 | $F_{52,1664}$=0.82 |
| Time | | p<0.0001 | $F_{52,1664}$=5.682 |
| Subregion | | p=0.6977 | $F_{1,32}$=0.1536 |
| *Figure 1—figure supplement 5Aiii* Habituation F-CS Responders | Two-way RM ANOVA | | |
| Time x Subregion | | p=0.0007 | $F_{52,884}$=1.782 |
| Time | | p<0.0001 | $F_{52,884}$=2.497 |
| Subregion | | p=0.7154 | $F_{1,17}$=0.1374 |
| *Figure 1—figure supplement 5Bii* R-US | Two-way RM ANOVA | | |
| Time x Subregion | | p=0.0002 | $F_{52,14248}$=1.87 |
| Time | | p=0.9740 | $F_{52,14248}$=0.66 |
| Subregion | | p=0.0021 | $F_{1,274}$=9.62 |
| *Figure 1—figure supplement 5Biv* F-US | Two-way RM ANOVA | | |
| Time x Subregion | | p<0.0001 | $F_{50,7100}$=2.29 |
| Time | | p<0.0001 | $F_{50,7100}$=5.03 |
| Subregion | | p=0.0587 | $F_{1,142}$=3.63 |
| *Figure 1—figure supplement 5Ci* Recall R-CS | Two-way RM ANOVA | | |
| Time x Subregion | | p=0.0537 | $F_{52,14560}$=1.34 |
| Time | | p=0.1030 | $F_{52,14560}$=1.26 |
| Subregion | | p=0.79 | $F_{1,280}$=0.07 |
| *Figure 1—figure supplement 5Ci* Recall F-CS | Two-way RM ANOVA | | |
| Time x Subregion | | p=0.5919 | $F_{52,14560}$=0.94 |
| Time | | p<0.0001 | $F_{52,14560}$=2.42 |
| Subregion | | p=0.3279 | $F_{1,280}$=0.961 |
| *Figure 1—figure supplement 5Ciii* Recall R-CS Responders | Two-way RM ANOVA | | |
| Time x Subregion | | p=0.1447 | $F_{52,1924}$=1.212 |
| Time | | p<0.0001 | $F_{52,1924}$=2.984 |
| Subregion | | p=0.1309 | $F_{1,37}$=2.387 |
| *Figure 1—figure supplement 5Ciii* Recall F-CS Responders | Two-way RM ANOVA | | |
| Time x Subregion | | p<0.0001 | $F_{52,1716}$=2.159 |
| Time | | p<0.0001 | $F_{52,1716}$=6.107 |
| Subregion | | p=0.4025 | $F_{1,33}$=0.7194 |
| *Figure 1—figure supplement 6Ai* Habituation R-CS | Two-way RM ANOVA | | |
| Time x Population | | p<0.0001 | $F_{64,3232}$=2.76 |
| Time | | p=0.1186 | $F_{32,3232}$=1.30 |
| Population | | p=0.9656 | $F_{2,101}$=0.036 |

*Appendix 1—table 1 continued*

| Dataset | Statistical test | p-values | F ratio (DFn, DFd) |
|---|---|---|---|
| *Figure 1—figure supplement 6Ai*<br>Habituation F-CS<br>Stage x Population<br>Stage<br>Population | Two-way RM ANOVA | <br><br>p<0.0001<br>p=0.0104<br>p=0.1205 | <br><br>$F_{64,4032}$=2.43<br>$F_{32,4032}$=1.67<br>$F_{2,126}$=2.15 |
| *Figure 1—figure supplement 6Aiii*<br>Habituation R-CS Responders<br>Time x Population<br>Time<br>Subregion | Two-way RM ANOVA | <br><br>p=0.9238<br>p<0.0001<br>p=0.9813 | <br><br>$F_{64,800}$=0.7528<br>$F_{32,800}$=7.436<br>$F_{2,25}$=0.0189 |
| *Figure 1—figure supplement 6Aiii*<br>Habituation F-CS Responders<br>Time x Population<br>Time<br>Subregion | Two-way RM ANOVA | <br><br>p=0.0005<br>p<0.0001<br>p=0.0473 | <br><br>$F_{64,1056}$=1.721<br>$F_{32,1056}$=2.425<br>$F_{2,33}$=3.353 |
| *Figure 1—figure supplement 6Bii*<br>R-US<br>Time x Population<br>Time<br>Population | Two-way RM ANOVA | <br><br>p<0.0001<br>p<0.0001<br>p<0.0001 | <br><br>$F_{56,3976}$=2.45<br>$F_{28,3976}$=9.61<br>$F_{2,142}$=11.18 |
| *Figure 1—figure supplement 6Biv*<br>F-US<br>Time x Population<br>Time<br>Population | Two-way RM ANOVA | <br><br>p<0.0001<br>p<0.0001<br>p=0.6376 | <br><br>$F_{56,4480}$=4.22<br>$F_{28,4480}$=31.28<br>$F_{2,160}$=0.451 |
| *Figure 1—figure supplement 6Ci*<br>Recall R-CS<br>Time x Population<br>Time<br>Population | Two-way RM ANOVA | <br><br>p<0.0001<br>p=0.5507<br>p=0.8385 | <br><br>$F_{72,9684}$=2.81<br>$F_{36,9684}$=0.952<br>$F_{2,269}$=0.176 |
| *Figure 1—figure supplement 6Ci*<br>Recall F-CS<br>Time x Population<br>Time<br>Population | Two-way RM ANOVA | <br><br>p<0.0001<br>p<0.0001<br>p=0.2932 | <br><br>$F_{72,9072}$=2.01<br>$F_{36,9072}$=9.42<br>$F_{2,252}$=1.23 |
| *Figure 1—figure supplement 6Ciii*<br>Recall R-CS Responders<br>Time x Population<br>Time<br>Population | Two-way RM ANOVA | <br><br>p<0.0001<br>p<0.0001<br>p=0.6363 | <br><br>$F_{72,2952}$=3.577<br>$F_{36,2952}$=16.4<br>$F_{2,82}$=0.4546 |
| *Figure 1—figure supplement 6Ciii*<br>Recall F-CS Responders<br>Time x Population<br>Time<br>Subregion | Two-way RM ANOVA | p<0.0001<br>p<0.0001<br>p=0.4251 | <br><br>$F_{72,3240}$=2.273<br>$F_{52,3240}$=28.5<br>$F_{2,90}$=0.8636 |
| *Figure 1—figure supplement 7Ai*<br>R-CS IC<br>Stage x Subregion<br>Stage<br>Subregion | Two-way ANOVA | <br><br>p<0.0001<br>p<0.0001<br>p<0.0001 | <br><br>$F_{9,3184}$=5.68<br>$F_{3,3184}$=9.99<br>$F_{3,3184}$=78.65 |
| *Figure 1—figure supplement 7Aii*<br>F-CS IC<br>Stage x Subregion<br>Stage<br>Subregion | Two-way ANOVA | <br><br>p<0.0001<br>p<0.0001<br>p<0.0001 | <br><br>$F_{9,3184}$=11.50<br>$F_{3,3184}$=22.38<br>$F_{3,3184}$=137.80 |
| *Figure 1—figure supplement 7Aiii*<br>R-CS CE<br>Stage x Population<br>Stage<br>Population | Two-way ANOVA | <br><br>p<0.0001<br>p<0.0001<br>p<0.0001 | <br><br>$F_{15,4776}$=29.17<br>$F_{3,4776}$=23.21<br>$F_{5,4776}$=28.00 |

*Appendix 1—table 1 continued*

| Dataset | Statistical test | p-values | F ratio (DFn, DFd) |
|---|---|---|---|
| ***Figure 1—figure supplement 7Aiv***<br>F-CS CE | Two-way ANOVA | | |
| Stage x Population | | p<0.0001 | $F_{15,4776}=7.93$ |
| Stage | | p<0.0001 | $F_{3,4776}=13.56$ |
| Population | | p<0.0001 | $F_{5,4776}=24.62$ |
| ***Figure 1—figure supplement 7Bi***<br>CS discrimination IC | Two-way ANOVA | | |
| Stage x Subregion | | p<0.0001 | $F_{3,1592}=18.53$ |
| Stage | | p<0.0001 | $F_{1,1592}=27.94$ |
| Subregion | | p<0.0001 | $F_{3,1592}=479.8$ |
| ***Figure 1—figure supplement 7Bii***<br>CS discrimination CE | Two-way ANOVA | | |
| Stage x Population | | p<0.0001 | $F_{5,2366}=15.58$ |
| Stage | | p<0.0001 | $F_{1,2366}=19.37$ |
| Population | | p<0.0001 | $F_{5,2366}=25.30$ |
| ***Figure 1—figure supplement 7Ci***<br>R-US IC | Two-way ANOVA | | |
| Stage x Subregion | | p<0.0001 | $F_{3,1592}=16.06$ |
| Stage | | p=0.0030 | $F_{1,1592}=8.85$ |
| Subregion | | p<0.0001 | $F_{3,1592}=302.70$ |
| ***Figure 1—figure supplement 7Ci***<br>F-US IC | Two-way ANOVA | | |
| Stage x Subregion | | p=0.2300 | $F_{3,1592}=1.44$ |
| Stage | | p=0.8792 | $F_{1,1592}=0.02$ |
| Subregion | | p<0.0001 | $F_{3,1592}=1619.00$ |
| ***Figure 1—figure supplement 7Ci***<br>R-US CE | Two-way ANOVA | | |
| Stage x Population | | p<0.0001 | $F_{5,2228}=5.40$ |
| Stage | | p=0.0213 | $F_{1,2228}=5.31$ |
| Subregion | | p<0.0001 | $F_{5,2228}=23.34$ |
| ***Figure 1—figure supplement 7Ci***<br>F-US CE | Two-way ANOVA | | |
| Stage x Population | | p=0.0013 | $F_{5,2388}=4.01$ |
| Stage | | p=0.0020 | $F_{1,2388}=9.53$ |
| Subregion | | p<0.0001 | $F_{5,2388}=76.81$ |
| ***Figure 1—figure supplement 7Cii***<br>RC CS←US transfer IC | Two-way ANOVA | | |
| Stage x Subregion | | p<0.0001 | $F_{9,3192}=242.50$ |
| Stage | | p<0.0001 | $F_{3,3192}=521.60$ |
| Subregion | | p<0.0001 | $F_{3,3192}=167.50$ |
| ***Figure 1—figure supplement 7Cii***<br>FC CS←US transfer IC | Two-way ANOVA | | |
| Stage x Subregion | | p<0.0001 | $F_{9,3192}=9.97$ |
| Stage | | p<0.0001 | $F_{3,3192}=11.62$ |
| Subregion | | p<0.0001 | $F_{3,3192}=688.50$ |
| ***Figure 1—figure supplement 7Cii***<br>RC CS←US transfer CE | Two-way ANOVA | | |
| Stage x Population | | p<0.0001 | $F_{5,4788}=18.39$ |
| Stage | | p=0.0013 | $F_{1,4788}=10.42$ |
| Population | | p<0.0001 | $F_{5,4788}=326.10$ |
| ***Figure 1—figure supplement 7Cii***<br>FC CS←US transfer CE | Two-way ANOVA | | |
| Stage x Population | | p<0.0001 | $F_{5,4788}=603.30$ |
| Stage | | p<0.0001 | $F_{1,4788}=142.20$ |
| Population | | p<0.0001 | $F_{5,4788}=1057.40$ |
| ***Figure 1—figure supplement 8Bi*** aIC R-US | Two-way RM ANOVA | | |
| Time x Performance | | p=0.0029 | $F_{52,7592}=1.63$ |
| Time | | p=0.0034 | $F_{52,7592}=1.62$ |
| Performance | | p=0.0388 | $F_{1,146}=4.35$ |

*Appendix 1—table 1 continued*

| Dataset | Statistical test | p-values | F ratio (DFn, DFd) |
|---|---|---|---|
| *Figure 1—figure supplement 8Bii* aIC F-US<br>Time x Performance<br>Time<br>Performance | Two-way RM ANOVA | <br>p<0.0001<br>p<0.0001<br>p=0.9474 | <br>$F_{50,3950}=2.47$<br>$F_{50,3950}=2.50$<br>$F_{1,79}=0.004$ |
| *Figure 1—figure supplement 8Biii* pIC R-US<br>Time x Performance<br>Time<br>Performance | Two-way RM ANOVA | <br>p<0.0001<br>p<0.0001<br>p=0.6200 | <br>$F_{52,6552}=9.86$<br>$F_{52,6552}=3.55$<br>$F_{1,126}=0.247$ |
| *Figure 1—figure supplement 8Biv* pIC F-US<br>Time x Performance<br>Time<br>Performance | Two-way RM ANOVA | <br>p<0.0001<br>p<0.0001<br>p<0.0001 | <br>$F_{50,3050}=3.27$<br>$F_{50,3050}=5.93$<br>$F_{1,61}=17.59$ |
| *Figure 1—figure supplement 8Ci* aIC R-CS<br>Time x Performance<br>Time<br>Performance | Two-way RM ANOVA | <br>p<0.0001<br>p=0.6431<br>p=0.5547 | <br>$F_{52,8268}=2.09$<br>$F_{52,8268}=0.917$<br>$F_{1,159}=0.350$ |
| *Figure 1—figure supplement 8Cii* aIC F-CS<br>Time x Performance<br>Time<br>Performance | Two-way RM ANOVA | <br>p=0.0434<br>p=0.0002<br>p=0.8521 | <br>$F_{52,8268}=1.36$<br>$F_{52,8268}=1.84$<br>$F_{1,159}=0.0349$ |
| *Figure 1—figure supplement 8Ciii* pIC R-CS<br>Time x Performance<br>Time<br>Performance | Two-way RM ANOVA | <br>p>0.9999<br>p=0.1204<br>p=0.8266 | <br>$F_{52,6188}=0.353$<br>$F_{52,6188}=1.24$<br>$F_{1,119}=0.05$ |
| *Figure 1—figure supplement 8Civ* pIC F-CS<br>Time x Performance<br>Time<br>Performance | Two-way RM ANOVA | <br>p=0.0003<br>p=0.0063<br>p=0.3331 | <br>$F_{52,6188}=1.81$<br>$F_{52,6188}=1.56$<br>$F_{1,119}=0.944$ |
| *Figure 1—figure supplement 8Di*<br>SST R-US<br>Time x Performance<br>Time<br>Performance | Two-way RM ANOVA | <br><br>p<0.0001<br>p<0.0001<br>p=0.0607 | <br><br>$F_{28,3052}=2.33$<br>$F_{28,3052}=2.49$<br>$F_{1,109}=3.59$ |
| *Figure 1—figure supplement 8Dii*<br>SST water<br>Time x Performance<br>Time<br>Performance | Two-way RM ANOVA | <br><br>p=0.1217<br>p<0.0001<br>p=0.0187 | <br><br>$F_{28,3052}=1.32$<br>$F_{28,3052}=20.42$<br>$F_{1,109}=5.70$ |
| *Figure 1—figure supplement 8Diii*<br>SST F-US<br>Time x Performance<br>Time<br>Performance | Two-way RM ANOVA | <br><br>p=0.0003<br>p<0.0001<br>p=0.9986 | <br><br>$F_{28,1344}=2.22$<br>$F_{28,1344}=8.10$<br>$F_{1,48}=3.012e-006$ |
| *Figure 1—figure supplement 8Div*<br>PKCδ R-US<br>Time x Performance<br>Time<br>Performance | Two-way RM ANOVA | <br><br>p<0.0001<br>p=0.0342<br>p=0.7613 | <br><br>$F_{28,3192}=3.07$<br>$F_{28,3192}=1.54$<br>$F_{1,114}=0.09$ |
| *Figure 1—figure supplement 8Dv*<br>PKCδ water<br>Time x Performance<br>Time<br>Performance | Two-way RM ANOVA | <br><br>p=0.0262<br>p<0.0001<br>p=0.6868 | <br><br>$F_{28,3220}=1.56$<br>$F_{28,3220}=2.67$<br>$F_{1,115}=0.16$ |
| *Figure 1—figure supplement 8Dvi*<br>PKCδ F-US<br>Time x Performance<br>Time<br>Performance | Two-way RM ANOVA | <br><br>p<0.0001<br>p<0.0001<br>p=0.3513 | <br><br>$F_{28,1848}=2.96$<br>$F_{28,1848}=6.32$<br>$F_{1,66}=0.88$ |

*Appendix 1—table 1 continued*

| Dataset | Statistical test | p-values | F ratio (DFn, DFd) |
|---|---|---|---|
| *Figure 1—figure supplement 8Dvii*<br>CEm R-US | Two-way RM ANOVA | | |
| Time x Performance | | p>0.999 | $F_{28,1512}$=0.18 |
| Time | | p<0.0001 | $F_{28,1512}$=3.95 |
| Performance | | p=0.7647 | $F_{1,54}$=0.09 |
| *Figure 1—figure supplement 8Dviii*<br>CEm water | Two-way RM ANOVA | | |
| Time x Performance | | p<0.0001 | $F_{28,1512}$=2.92 |
| Time | | p<0.0001 | $F_{28,1512}$=3.53 |
| Performance | | p=0.5895 | $F_{1,54}$=0.30 |
| *Figure 1—figure supplement 8Dix*<br>CEm F-US | Two-way RM ANOVA | | |
| Time x Performance | | p=0.9646 | $F_{28,1204}$=0.57 |
| Time | | p<0.0001 | $F_{28,1204}$=22.73 |
| Performance | | p=0.6271 | $F_{1,43}$=0.24 |
| *Figure 2—figure supplement 2Ai*<br>Feature importance behavior | One-way ANOVA | p<0.0001 | $F_{9,156190}$=5184.00 |
| *Figure 2—figure supplement 2Aii*<br>Feature importance CS | One-way ANOVA | p<0.0001 | $F_{9,73454}$=2437.00 |
| *Figure 2—figure supplement 2Aiii*<br>Feature importance CS hM4 CE$^{PKC\delta}$ | One-way ANOVA | p<0.0001 | $F_{9,77790}$=5185.00 |
| *Figure 2—figure supplement 2Aiv*<br>Feature importance US hM4 CE$^{PKC\delta}$ | One-way ANOVA | p<0.0001 | $F_{9,77821}$=2126.00 |
| *Figure 2—figure supplement 2Av*<br>Feature importance habituation CS hM4 CE$^{PKC\delta}$ | One-way ANOVA | p<0.0001 | $F_{9,77619}$=1766.00 |
| *Figure 2—figure supplement 5Ai*<br>RC port visits opto | Two-way RM ANOVA | | |
| Session x Treatment | | p<0.0001 | $F_{14,154}$=3.40 |
| Session | | p<0.0001 | $F_{7,154}$=15.39 |
| Treatment | | p=0.0271 | $F_{2,22}$=4.27 |
| *Figure 2—figure supplement 5Aii*<br>FC freezing opto | Two-way RM ANOVA | | |
| Treatment x Trial | | p<0.0001 | $F_{10,110}$=4.71 |
| Treatment | | p<0.0001 | $F_{5,110}$=50.67 |
| Trial | | p=0.0054 | $F_{2,22}$=6.69 |
| *Figure 2—figure supplement 5Bi*<br>Recall port visits opto | Two-way RM ANOVA | | |
| CS x Treatment | | p=0.2605 | $F_{4,44}$=1.37 |
| CS | | p<0.0001 | $F_{2,44}$=17.90 |
| Treatment | | p=0.0343 | $F_{2,22}$=3.95 |
| *Figure 2—figure supplement 5Bii*<br>Recall freezing opto | Two-way RM ANOVA | | |
| CS x Treatment | | p=0.0225 | $F_{4,44}$=3.17 |
| CS | | p<0.0001 | $F_{2,44}$=66.66 |
| Treatment | | p=0.0154 | $F_{2,22}$=5.07 |
| *Figure 2—figure supplement 7Ai*<br>RC GFP | Two-way RM ANOVA | | |
| CS x Session | | p<0.0001 | $F_{13,130}$=5.97 |
| CS | | p=0.0001 | $F_{1,10}$=38.23 |
| Session | | p<0.0001 | $F_{13,130}$=4.64 |
| *Figure 2—figure supplement 7A* ii<br>RC hM4(pIC)-CE | Two-way RM ANOVA | | |
| CS x Session | | p<0.0001 | $F_{13,91}$=9.07 |
| CS | | p=0.0027 | $F_{1,7}$=20.62 |
| Session | | p<0.0001 | $F_{13,91}$=8.09 |
| *Figure 2—figure supplement 7Aiii*<br>RC hM4(aIC)-pIC | Two-way RM ANOVA | | |
| CS x Session | | p=0.0003 | $F_{13,104}$=3.30 |
| CS | | p=0.0053 | $F_{1,8}$=14.36 |
| Session | | p<0.0001 | $F_{13,104}$=4.86 |

*Appendix 1—table 1 continued*

| Dataset | Statistical test | p-values | F ratio (DFn, DFd) |
|---|---|---|---|
| *Figure 2—figure supplement 7Aiv*<br>FC | Two-way RM ANOVA | | |
| Treatment x Trial | | p=0.0537 | $F_{10,125}$=1.88 |
| Treatment | | p=0.56 | $F_{2,25}$=0.59 |
| Trial | | p<0.0001 | $F_{5,125}$=174.70 |
| *Figure 2—figure supplement 7B*<br>Recall port visits hM4(pIC)-CE | Two-way RM ANOVA | | |
| CS x Treatment | | p=0.9251 | $F_{2,36}$=0.08 |
| CS | | p<0.0001 | $F_{2,36}$=31.88 |
| Treatment | | p=0.7647 | $F_{1,18}$=0.09 |
| *Figure 2—figure supplement 7B*<br>Recall freezing hM4(pIC)-CE | Two-way RM ANOVA | | |
| CS x Treatment | | p=0.2970 | $F_{2,36}$=1.26 |
| CS | | p<0.0001 | $F_{2,36}$=40.47 |
| Treatment | | p=0.0087 | $F_{1,18}$=8.65 |
| *Figure 2—figure supplement 7B*<br>Recall port visits hM4(aIC)-pIC | Two-way RM ANOVA | | |
| CS x Treatment | | p=0.9652 | $F_{2,38}$=0.04 |
| CS | | p<0.0001 | $F_{2,38}$=30.32 |
| Treatment | | p=0.7341 | $F_{1,19}$=0.19 |
| *Figure 2—figure supplement 7B*<br>Recall freezing hM4(aIC)-pIC | Two-way RM ANOVA | | |
| CS x Treatment | | p=0.4522 | $F_{2,38}$=0.81 |
| CS | | p<0.0001 | $F_{2,38}$=54.69 |
| Treatment | | p=0.0008 | $F_{1,19}$=15.80 |
| *Figure 5—figure supplement 1A*<br>RF Da Control | One-way ANOVA | p<0.0001 | $F_{2,5997}$=271.60 |
| *Figure 5—figure supplement 3Ai*<br>RC Control | Two-way RM ANOVA | | |
| CS x Session | | p=0.0163 | $F_{9,54}$=2.54 |
| CS | | p=0.0745 | $F_{1,6}$=4.65 |
| Session | | p=0.1830 | $F_{9,54}$=1.47 |
| *Figure 5—figure supplement 3Aii*<br>RC CE$^{PKC\delta}$-NBM | Two-way RM ANOVA | | |
| CS x Session | | p=0.9163 | $F_{9,45}$=0.42 |
| CS | | p=0.1080 | $F_{1,5}$=3.82 |
| Session | | p=0.5668 | $F_{9,45}$=0.86 |
| *Figure 5—figure supplement 3Aiii*<br>FC freezing | Two-way RM ANOVA | | |
| Trial x Treatment | | p=0.5363 | $F_{5,60}$=0.83 |
| Trial | | p<0.0001 | $F_{5,60}$=15.37 |
| Treatment | | p=0.7568 | $F_{1,12}$=0.10 |
| *Figure 5—figure supplement 3Bi*<br>Recall port visits CE$^{PKC\delta}$-NBM | Two-way RM ANOVA | | |
| CS x Treatment | | p=0.0338 | $F_{2,22}$=3.97 |
| CS | | p=0.0434 | $F_{2,22}$=3.63 |
| Treatment | | p=0.5666 | $F_{1,11}$=0.35 |
| *Figure 5—figure supplement 3Bii*<br>Recall freezing CE$^{PKC\delta}$-NBM | Two-way RM ANOVA | | |
| CS x Treatment | | p=0.0095 | $F_{2,22}$=3.11 |
| CS | | p<0.0001 | $F_{2,22}$=81.45 |
| Treatment | | p=0.0029 | $F_{1,11}$=8.72 |
| *Figure 5—figure supplement 3Ci*<br>RC Control | Two-way RM ANOVA | | |
| CS x Session | | p=0.0109 | $F_{9,72}$=2.63 |
| CS | | p=0.0104 | $F_{1,8}$=11.08 |
| Session | | p=0.0048 | $F_{9,72}$=2.96 |
| *Figure 5—figure supplement 3Cii*<br>RC CE$^{SST}$-NBM | Two-way RM ANOVA | | |
| CS x Session | | p<0.0001 | $F_{9,108}$=4.75 |
| CS | | p=0.0062 | $F_{1,12}$=10.99 |
| Session | | p=0.0008 | $F_{9,108}$=3.48 |

*Appendix 1—table 1 continued*

| Dataset | Statistical test | p-values | F ratio (DFn, DFd) |
|---|---|---|---|
| *Figure 5—figure supplement 3Ciii* <br> FC freezing | Two-way RM ANOVA | | |
| Trial x Treatment | | p=0.3490 | $F_{5,100}=1.13$ |
| Trial | | p<0.0001 | $F_{5,100}=119.10$ |
| Treatment | | p=0.0038 | $F_{1,20}=10.70$ |
| *Figure 5—figure supplement 3Di* <br> Recall port visits $CE^{SST}$-NBM | Two-way RM ANOVA | | |
| CS x Treatment | | p=0.9034 | $F_{2,40}=0.10$ |
| CS | | p=0.0001 | $F_{2,40}=11.59$ |
| Treatment | | p=0.5441 | $F_{1,20}=0.38$ |
| *Figure 5—figure supplement 3Dii* <br> Recall freezing $CE^{SST}$-NBM | Two-way RM ANOVA | | |
| CS x Treatment | | p=0.9419 | $F_{2,40}=0.06$ |
| CS | | p<0.0001 | $F_{2,40}=77.68$ |
| Treatment | | p=0.5966 | $F_{1,20}=0.29$ |
| *Figure 5—figure supplement 5Ai* <br> CS CE | Two-way ANOVA | | |
| Stage x Population | | p<0.0001 | $F_{15,9576}=34.91$ |
| Stage | | p<0.0001 | $F_{3,9576}=89.24$ |
| Population | | p<0.0001 | $F_{5,9576}=328.04$ |
| *Figure 5—figure supplement 5Aii* <br> CS similarity CE | Two-way ANOVA | | |
| Stage x Population | | p<0.0001 | $F_{5,9588}=311.70$ |
| Stage | | p<0.0001 | $F_{1,9588}=221.90$ |
| Population | | p<0.0001 | $F_{5,9588}=110.60$ |
| *Figure 5—figure supplement 5Aiii* <br> CS←US transfer CE | Two-way ANOVA | | |
| Stage x Population | | p<0.0001 | $F_{5,9588}=163.90$ |
| Stage | | p<0.0001 | $F_{1,9588}=432.50$ |
| Population | | p<0.0001 | $F_{5,9588}=584.50$ |
| *Figure 5—figure supplement 5B* <br> CS discrimination CE | Two-way ANOVA | | |
| Treatment x Population | | p<0.0001 | $F_{5,2387}=29.37$ |
| Treatment | | p<0.0001 | $F_{1,2387}=116.3$ |
| Population | | p<0.0001 | $F_{5,2387}=130.1$ |
| *Figure 5—figure supplement 5C* <br> RC CS←US transfer CE hM4(aIC) | Two-way ANOVA | | |
| Treatment x Population | | p<0.0001 | $F_{5,4788}=186.13$ |
| Treatment | | p<0.0001 | $F_{1,4788}=259.70$ |
| Population | | p<0.0001 | $F_{5,4788}=64.77$ |
| *Figure 5—figure supplement 5C* <br> FC CS←US transfer CE hM4(aIC) | Two-way ANOVA | | |
| Treatment x Population | | p<0.0001 | $F_{5,4788}=195.10$ |
| Treatment | | p<0.0001 | $F_{1,4788}=481.90$ |
| Population | | p<0.0001 | $F_{5,4788}=948.20$ |

DFn = degrees of freedom for numerator, DFd = degrees of freedom for denominator.

