## [Decision Letter]

**Acceptance summary:**

The neural correlates of stimulus features are distributed across the brain. Using a powerful combination of techniques, authors uncovered a mechanism for associative learning organized by the processing of stimulus features such as value, salience, and valence, across a cortico-limbic network. They demonstrate a new link between interoceptive efferents and affective learning.

**Decision letter after peer review:**

Thank you for submitting your article "The amygdala instructs insular feedback for affective learning" for consideration by *eLife*. Your article has been reviewed by three peer reviewers, one of whom is a member of our Board of Reviewing Editors, and the evaluation has been overseen by Kate Wassum as the Senior Editor.

The reviewers have discussed the reviews with one another and the Reviewing Editor has drafted this decision to help you prepare a revised submission.

Summary:

In this study by Kargl et al., authors use a powerful combination of techniques (chemogenetics, optogenetics, calcium imaging, electrophysiology, and fMRI) in mice to probe the roles of insular cortex (IC), central amygdala (CE), and basal forebrain (BF) in Pavlovian conditioning. Authors use a behaviorally straightforward paradigm to interrogate conditioned responding to a CS, and specifically features of CS and US value, salience, and valence. Authors find evidence for a hierarchical network in reward and threat learning, such that 1) IC-CE are functionally-coupled and facilitate conditioned responding, 2) acquisition establishes an intra-cortical hierarchy (with aIC above pIC), 3) CS salience in the CE recruits CS representations bottom-up via BF and 4) Ce-I incorporates feedback to discriminate CS valence, top-down. Results are interesting and novel. However, essential revisions were identified as largely limited to issues of clarity and data presentation

Essential Revisions:

1) There were concerns in particular about the density of the data shown in figures which are drawn from very diverse approaches, making the conclusions difficult to follow as there is frequent cross-reference to supplementary figures/data. The primary suggestion is to distil the data and streamline presentation for a clearer, easier-to-follow reading. In particular, authors are encouraged to quantitatively and convincingly show how the conclusion of a functional network between IC, CE and BF was extracted from the RS-fMRI data. The data now shown in Figure 1A and Figure 1—figure supplement 1 are difficult to follow and should be further described, since they constitute a major novelty of the report.

2) Relatedly, the manuscript would benefit from more explanation and rationale for the many technologies and analyses, and by first showing raw data before classifier results. In particular, Figures 1Cii and Dii were difficult to follow and unconvincing: while the authors show that the US and CS become more similar for the classifier, it is unclear why this is the case. For example, could it be general arousal, salience, or bodily responses detected in the IC? Alternatively, the authors could train a decoder to discriminate the two CSs (R-CS and F-CS), instead of training classifiers to decode R-CS or F-CS from the other CS. It is recommended to either exclude or reanalyze these data.

3) There could be more detail about experimental design before the Materials and methods, which unfortunately reveals important information rather late in the manuscript. For example, it could be mentioned more clearly that circuit manipulations were conducted/replicated using different technologies in separate experiments and in separate cohorts of animals. Before this statement, it is somewhat unclear if the decoding was conducted in real-time, and if (CE-NBM) stimulation during in vivo recording or if optogenetic manipulation occurred in closed-loop.

4) Authors should include animal numbers throughout, instead of general ranges in the Data analyses and statistical tests section.

5) Relevant statistical analyses must be included within the manuscript itself.

---

## [Author Response]

Essential Revisions:1) There were concerns in particular about the density of the data shown in figures which are drawn from very diverse approaches, making the conclusions difficult to follow as there is frequent cross-reference to supplementary figures/data. The primary suggestion is to distil the data and streamline presentation for a clearer, easier-to-follow reading.

We thank the reviewers for this suggestion. The rationale for the current arrangement of data was to group similar data types within single supplementary figures, irrespective of their sequence in the manuscript, to avoid mixing experimental approaches.

To address this while keeping this overall flow, we have edited the text throughout for better clarity. Further, we now better explain the technologies and methods used, and have expanded on the rationale for our experimental design (see also Essential Revision #2).

In particular, authors are encouraged to quantitatively and convincingly show how the conclusion of a functional network between IC, CE and BF was extracted from the RS-fMRI data. The data now shown in Figure 1A and Figure 1—figure supplement 1 are difficult to follow and should be further described, since they constitute a major novelty of the report.

This is an excellent suggestion. We complemented the voxel/seed-based slice view of fMRI data in Figure 1A with region-based functional connectivity from the ROI correlation matrix in Figure 1—figure supplement 1B. IC subregions in the correlation matrix (‘agranular area’, ‘gustatory area’, ‘visceral area’) were summarized into a single IC area for each hemisphere and correlations were re-computed. The resulting networks were added as Figure 1B/Figure 1—figure supplement 1C. To avoid overcrowding, only areas significantly correlating with IC or CE are shown as nodes. We believe these data now conclusively identify the IC-CE/NBM network as a distinct functional unit, and that this is presented in a more accessible manner.

2) Relatedly, the manuscript would benefit from more explanation and rationale for the many technologies and analyses, and by first showing raw data before classifier results.

We thank the reviewers for this suggestion. We added explanatory sentences in the main text (subsection “IC and CE are functionally coupled and acquire CS information”) before introducing new analyses. We have also referred to raw electrophysiological and Ca^2+^ imaging data in Figure 1—figure supplements 5, 6 prior to the introduction of decoding approaches in Figure 1.

In particular, Figures 1Cii and Dii were difficult to follow and unconvincing: while the authors show that the US and CS become more similar for the classifier, it is unclear why this is the case. For example, could it be general arousal, salience, or bodily responses detected in the IC?

We thank the reviewers for the suggestion. As noted by the reviewer, CS and US become more similar when using similarity decoding (Figure 1Ciii/Diii). In this context, these figures are intended to demonstrate exactly what the reviewer noted: the CS incorporating general arousal, salience, or bodily responses from US representations, as shown in Figure 1Ciii/Diii. We believe that our interpretation stands and have emphasized this point in the text (subsection “IC and CE are functionally coupled and acquire CS information”).

Alternatively, the authors could train a decoder to discriminate the two CSs (R-CS and F-CS), instead of training classifiers to decode R-CS or F-CS from the other CS. It is recommended to either exclude or reanalyze these data.

We thank the reviewers for the suggestion to clarify and expand on our decoding approach. In addition to training classifiers to decode R-CS or F-CS from the other CS, we have added an alternative decoding approach that attempts to directly discriminate R-CS and F-CS. This has been included as Figure 1—figure supplement 7Bi, ii and Figure 5—figure supplement 5B, which complement Figure 1Cii, 1Dii and Figure 5Eiii, respectively.

With these complementary decoding approaches, we could differentiate between two important aspects of stimulus-related information. A decoder trained to discriminate R-CS and F-CS is sensitive to differences in (primary) sensory representations. Decoding R-CS or F-CS from the other CS explicitly reports shared features among CS representations. We have clarified this point in the manuscript (subsection “IC and CE are functionally coupled and acquire CS information”).

3) There could be more detail about experimental design before the Materials and methods, which unfortunately reveals important information rather late in the manuscript. For example, it could be mentioned more clearly that circuit manipulations were conducted/replicated using different technologies in separate experiments and in separate cohorts of animals. Before this statement, it is somewhat unclear if the decoding was conducted in real-time, and if (CE-NBM) stimulation during in vivo recording or if optogenetic manipulation occurred in closed-loop.

We thank the reviewers for the suggestion. The respective sections have been edited for clarity, e.g. the experimental approach for the chemogenetic manipulation of pIC–CE circuitry has been introduced prior to the results (subsection “IC–CE information flow facilitates conditioned responding”). Also, we have explained the open-loop design of the CE-NBM stimulation during IC in vivo electrophysiological recordings (subsection “The basal forebrain mediates bottom-up recruitment of IC activity”).

4) Authors should include animal numbers throughout, instead of general ranges in the Data analyses and statistical tests section.

The number of animals used for each experiment are included in figure legends.

5) Relevant statistical analyses must be included within the manuscript itself.

Statistical analyses are now integrated into the figure legends in addition to Appendix 1—table 1.